# Rapid detection and capture of clinical *Escherichia coli* strains mediated by OmpA-targeting nanobodies
Michèle Sorgenfrei [1], Lea M. Hürlimann[1], Andrea Printz [1,2], Fanny Wegner[1], Damien Morger[3], Fabian Ackle [1], Mélissa M. Remy[2], Grzegorz Montowski [3], Hans-Anton Keserue[3], Aline Cuénod [1], Frank Imkamp[1], Adrian Egli[1], Peter M. Keller[2,4] & Markus A. Seeger [1] ✉

*Escherichia coli* is a major cause of blood stream and urinary tract infections. Owing to the spread of antimicrobial resistance, it is often treated with an inadequate antibiotic. With the aim to accelerate the diagnostics of this key pathogen, we used the flycode technology to generate nanobodies against the conserved and highly abundant outer membrane protein OmpA. Two nanobodies each recognizing a different isoform of OmpA were shown by flow cytometry to recognize > 91% of 85,680 *E. coli* OmpA sequences deposited in a large bacterial genome database. Crystal structures of these nanobodies in complex with the respective OmpA isoform revealed interactions with all four surface accessible loops of OmpA. Steric hindrance caused by dense O-antigen layers initially impeded reliable capture of clinical *E. coli* strains. By generating nanobody constructs with long linkers and by thinning the O-antigen layer through alterations to growth medium and buffers, we achieved to capture < 50 CFU/mL. Our work provides a framework to generate nanobodies for the specific and sensitive detection and capture of clinically relevant pathogenic bacteria.

*Escherichia coli* (*E. coli*) is an abundant and highly diverse Gram-negative bacterial species that naturally inhabits the gut of humans and animals but is also ubiquitous in the environment. Despite being a commensal bacterium that is part of the gut microbiome, it is a leading cause for bloodstream infections, urinary tract infections, and severe intestinal diseases[1]. Antimicrobial resistance (AMR) is increasingly prevalent in pathogenic *E. coli* strains. In particular, resistance to cephalosporins and carbapenems caused by extended spectrum β-lactamases (ESBL) or carbapenemases are of high concern[2]. This explains why this species has been designated as a priority pathogen by the World Health Organization[3].

The clinical manifestation of *E. coli* can be highly diverse and is strain- and lineage dependent[4]. Accordingly, the species of *E. coli* is subdivided into eight "pathotypes", which fall into four main categories, namely enteric and diarrheal disease, urinary tract infections (UTIs) as well as bloodstream infections (BSIs) and meningitis[1]. Characteristic sets of virulence genes equip pathogenic bacteria to cause specific pathotypes. A hallmark of diarrhoeagenic *E. coli* strains is the secretion of toxins, which can be life-threatening[5]. Of note, genetic studies have clearly shown that *Shigella*, despite being historically assigned as a separate species, is a specialized diarrheagenic *E. coli*[4]. Based on large sets of genome data, *E. coli* has been classified into eight phylogroups[4,6]

and a more fine-grained analysis subdivides the species into 50 main lineages[4].

The *E. coli* species displays a large diversity of O-antigens that are part of the lipopolysaccharides (LPS) and exposed at the cell surface. Historically, different *E. coli* strains have been classified according to their serotype; that is a group of *E. coli* strains being recognized by the same polyclonal antibodies generated in mice or rabbits upon immunization with various strains. According to a recent review, there are 185 known serogroups known for *E. coli*, each having a different O-antigen structure[7]. The extraordinary O-antigen variability at the surface of *E. coli* complicates the use of antibodies for diagnostic purposes because an impractically large number of binders would be needed to achieve reasonable species coverage.

In contrast, outer membrane proteins (OMPs) are considerably more conserved in terms of their sequence. By occupying >50% of the outer membrane surface[8–10], OMPs are an integral part of the asymmetric LPS bilayer, and they are among the most highly expressed proteins in proteomes of Gram-negative bacteria. With an estimated 100,000 proteins per cell, OmpA is the most abundant OMP of *E. coli* and occupies around 6–20% of the bacterial surface alone, depending on the cell size[10]. However, in intact cells, the surface-accessible loops of OmpA (and of other OMPs) are shielded by dense LPS O-antigens, explaining the paucity of reported

[1]Institute of Medical Microbiology, University of Zurich, Zurich, Switzerland. [2]Institute for Infectious Diseases, University of Bern, Bern, Switzerland. [3]rqmicro AG, Schlieren, Switzerland. [4]Present address: Clinical Bacteriology / Mycology, University Hospital Basel, Basel, Switzerland. ✉e-mail: m.seeger@imm.uzh.ch

monoclonal antibodies against *E. coli* OmpA[11,12]. Of note, these previously identified OmpA antibodies have not been demonstrated to reliably recognize a larger set of clinical strains.

*E. coli* is a leading cause of bloodstream and urinary tract infections, and considering the increasing global AMR burden, novel diagnostic tests that would allow for the rapid detection of frequent resistance mechanisms are urgently needed to curb AMR. Especially bloodstream infections (BSIs) pose a diagnostic challenge. If treated with an ineffective antibiotic against which the infecting pathogen is resistant, the survival rate of BSI patients drops by around 8% every hour after the onset of recurrent or persistent hypotension, which marks the start of a septic shock[13]. Yet, in patients suffering from BSI, the bacterial load is estimated in the range of only 1–1000 cells per milliliter of blood, depending on whether the count is determined by conventional plating or the number of genome copies[14]. Therefore, blood culturing is required to first multiply the bacterial numbers for subsequent diagnostic analysis. For *E. coli*, the median time to positivity of blood cultures is around 11 h[15]. Hence, valuable time is lost while waiting for the blood culture to turn positive. This time window can be shortened if one develops diagnostic tools that allow for pathogen capture directly from patient samples or early blood cultures that have not yet turned positive.

There have been recent technical developments tackling the challenge of accelerating the diagnostic testing of BSIs using DNAzyme technology[16] or chemically functionalized magnetic nanoparticles for bacterial capture in a microfluidic devices[17,18]. However, when applying these new methods for diagnostic purposes, the bacterial cells are often lysed during the process of analysis, thereby impeding phenotypic antibiotic susceptibility testing (AST).

With the aim to accelerate diagnostics of *E. coli* infections using AST and to readily detect *E. coli* in environmental samples, we generated nanobodies targeting OmpA for the rapid staining and capture of viable pathogens. Nanobodies are the variable domain of heavy chain-only antibodies found in camelids such as alpacas[19], and can be generated against any protein including challenging membrane proteins[20,21]. A hallmark of nanobodies is their small size; with 15 kDa they are ~10 times smaller than classical antibodies, which predestines them to target partially hidden epitopes such as extracellular loops of OMPs embedded in the LPS layer of Gram-negative pathogens[22]. Owing to their high biophysical robustness and inexpensive production, nanobodies are ideally suited to develop economically viable diagnostic tests[23].

## Results

### Selection of OmpA as target protein
We identified OmpA as a suitable surface-accessible and abundantly expressed target to generate nanobodies for staining and capture of living *E. coli* cells. In addition, OmpA is non-essential, which is relevant for the live capture of *E. coli* for phenotypic antimicrobial susceptibility testing. OmpA has also been identified as a target for several bacteriophages[24–27] showing that it can be recognized by large molecules. To gain insights into the prevalence of OmpA in *E. coli* genomes, we searched for the *ompA* gene in a large database containing >661,000 bacterial genomes (henceforth called the 661k database)[28], which contains ~90,000 assembled *E. coli* genomes. Our bioinformatics analysis (see methods) identified the *ompA* gene in 95% of these *E. coli* genomes. When clustering the OmpA protein sequences phylogenetically (Supplementary Fig. 1), they separated into two main groups, corresponding to the two isoforms described earlier[29]. Separation according to the two isoforms also manifested when hierarchical clustering of the loop variants was performed [29] (Supplementary Fig. 2). In this study, we refer to these isoforms as OmpA-short and OmpA-long, because their extracellular loop 3 differs in length. Within these two isoforms, the OmpA sequences are highly conserved (Supplementary Figs. 1 and 2).

### Generation of nanobodies recognizing OmpA in the native context of *E. coli*
The most frequent OmpA-short and OmpA-long sequences (reference sequences in Supplementary Figs. 3 and 4 and Supplementary Table 1) were cloned, expressed, and purified to immunize two alpacas (see methods). After RNA extraction from lymphocytes, two consecutive rounds of phage display were carried out[21,30], followed by deep screening using the flycode technology[22] to identify nanobodies that bind OmpA in the native context of *E. coli*. Flycodes are genetically encoded peptide barcodes optimized for detection by mass spectrometry. Approximately 2000 nanobodies pre-enriched by phage display were tagged with flycodes, deep-sequenced and screened against different *E. coli* strains. Our strain set included the laboratory *E. coli* K-12 strain and its isogenic *ompA* knockout, both devoid of O-antigens, as well as six clinical *E. coli* isolates shielded by different O-antigens and expressing either the short or long OmpA isoform (Supplementary Table 1, Supplementary Fig. 3 and 4). Thereby, we identified two nanobodies, Nb01 and Nb39, which according to our quantitative flycode analysis, exhibit high binding affinity and specificity for OmpA-short or OmpA-long, respectively, in the cellular context of clinical *E. coli* strains (Supplementary Tables 2–6) (Fig. 1).

### Validation of Nb01 and Nb39 in cellular binding assays
Nb01 and Nb39 were cloned and purified as individual proteins and their ability to specifically bind to their respective OmpA isoforms was analyzed in a cellular binding assay using fluorescently labeled nanobodies against the same set of *E. coli* strains as used for flycode screening. For the cellular binding experiments, we either labeled the nanobody directly with AlexaFluor 647 (AF647) via a cysteine introduced at the C-terminus or indirectly via an NTA-biotin linker bound to Atto565-labeled streptavidin, which recognizes the C-terminal His-tag of the respective nanobody (Supplementary Fig. 5). Since *E. coli* MC1061 expresses OmpA-short, we complemented the *E. coli* MC1061 Δ*ompA* strain with a plasmid expressing OmpA-long to characterize Nb39. When directly labeled with AF647, both Nb01 and Nb39 recognized all *E. coli* strains including the clinical isolates expressing the respective OmpA isoform, showing that these nanobodies can stain a diverse set of strains (Fig. 2a, b). In contrast, when performing the same assay with indirect labeling via streptavidin, only the lab strain *E. coli* MC1061 devoid of O-antigen and clinical strain #1 (CS#1) were recognized (Supplementary Fig. 6). This shows that most clinical strains possess a dense O-antigen layer that prevents the bulky nanobody-streptavidin complex from binding to OmpA, thereby directly demonstrating the advantage of using small nanobodies for the staining and capture of clinical *E. coli* strains. Intriguingly, the nanobodies were found to be highly specific. Nb01 did not recognize any of the strains carrying the OmpA-long isoform, and conversely, Nb39 did not recognize strains expressing OmpA-short. *E. coli* MC1061 Δ*ompA* was not recognized by any of the nanobodies (Fig. 2b).

### Structural characterization of nanobodies
To gain insights into the binding geometry of Nb01 and Nb39 and to rationalize their specificity and species coverage, we determined crystal structures of Nb01-OmpA-short and Nb39-OmpA-long, both resolved at a resolution of around 2.3 Å (Fig. 3, Supplementary Table 7). The structures revealed that both nanobodies bind to the surface-accessible epitope of OmpA, consistent with their ability to recognize OmpA in the cellular context. A closer inspection of the binding interface by PDBePISA (https://www.ebi.ac.uk/pdbe/pisa/) revealed that the nanobodies interact with all four extracellular loops of OmpA (Fig. 3c and d), including extensive contacts with the variable loop 3, thereby explaining the high specificity for the respective isoform. Nb01 binds with all its three complementary determining regions (CDRs), whereas Nb39 interacts via CDR1 and CDR3 only. The complex interfaces are 548 Å$^2$ and 658 Å$^2$ for Nb01-OmpA-short and Nb39-OmpA-long, respectively. A recent analysis of a larger number of SARS-CoV-2 nanobodies and antibodies revealed that the buried surface areas of nanobodies range from 500 Å$^2$ to 1000 Å$^2$ with a median of around 800 Å$^2$ [31], indicating the binder interfaces of Nb01 and Nb39 are in the lower range of typical nanobody-target protein complexes. The same study revealed that the median buried surface area of antibodies is around 100 Å$^2$ larger than that of nanobodies.

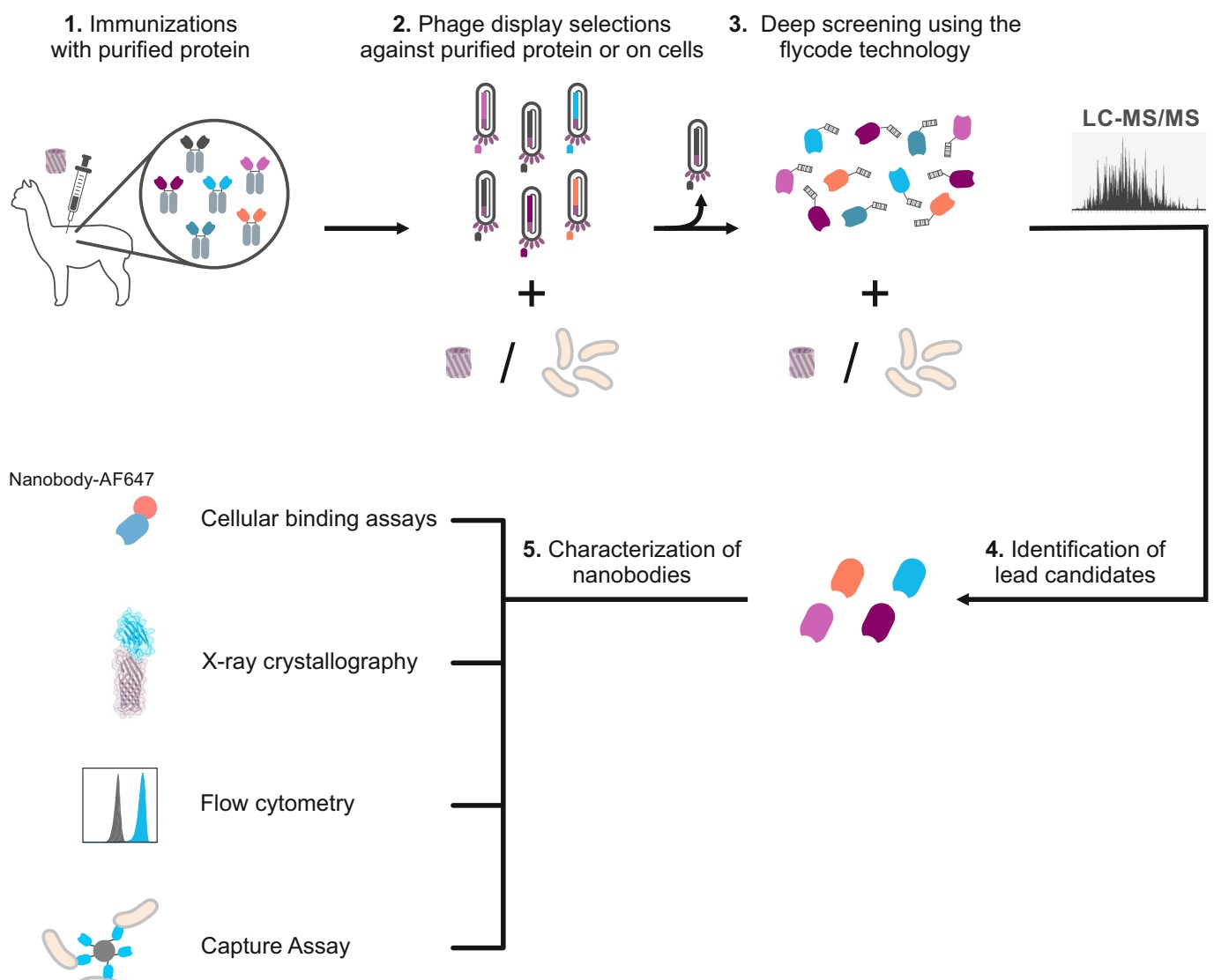

**Fig. 1 | Workflow for identification and characterization of nanobodies targeting OmpA in the cellular context of *E. coli*. 1** Detergent-purified OmpA was used to immunize alpacas, and a nanobody phage display library was generated. **2** Two rounds of phage display were performed to enrich nanobodies against purified OmpA or *E. coli* cells expressing OmpA on their surface. **3** Enriched nanobody sequences were cloned into a vector library containing unique peptide barcodes (flycodes) and a pool of nanobodies each containing a flycode was purified. **4** Using mass spectrometry (LC-MS/MS), flycodes allowed for the efficient identification of lead nanobodies which bind to OmpA in the native bacterial context. **5** Identified nanobodies were thoroughly characterized in cellular binding assays, flow cytometry using fluorescent labeling, X-ray crystallography, or capture assays with magnetic bead coupling.

## Sequence conservation analysis reveals high species coverage

To obtain detailed insights into the species coverage of the nanobodies, we performed a comprehensive sequence analysis of OmpA variants in clinical *E. coli* isolates. The analysis is based on two databases. The first database stems from the University Hospital Basel (USB) and the Institute of Medical Microbiology at the University of Zurich (IMM), representing the German-speaking part of Switzerland. It is hereafter named Swiss dataset and contains 2,034 *E. coli* strains encoding a complete *ompA* sequence (Fig. 2c). Importantly, we had physical access to these strains for experimental validation. The second source of sequences is the above mentioned 661k database (Fig. 2d)[28], which contains 85,680 *E. coli* strains having a complete *ompA* gene present.

We performed a protein sequence analysis of the extracellular loops (underlined sequences in Fig. 3c and d) and identified 32 unique OmpA loop combinations in the Swiss dataset (Fig. 2c), and 179 unique OmpA loop combinations in the 661 k database (Fig. 2d). By combining this sequence information with testing representative strains in a high-throughput

binding assay based on flow cytometry (Supplementary Fig. 7), we categorized the strains into three groups: recognized and thus covered by one of our two nanobodies (detected), not recognized (not detected), and unknown (when no representative strain was tested or available) (Supplementary Table 8). This analysis was conducted for both our Swiss dataset and the 661k dataset, respectively (Fig. 2c, d). Remarkably, with our two OmpA nanobodies, we achieved a species coverage of 79% of the total 2,034 available *E. coli* strains from the Swiss database and 91% of the 85,680 *E. coli* strains from the 661k database (Fig. 2e).

We identified five OmpA variants which collectively account for 97% of the non-detected strains of the Swiss database and 92% of the non-detected *E. coli* strains of the 661k database (largest five red bars in Fig. 2c, d respectively). For simplicity, we call these non-detected variants OmpA-ND#1-5. A closer inspection of the respective loop regions revealed that the deviating residues map to positions which interact with the nanobodies in the crystal structures (Fig. 3, Supplementary Fig. 8). For example, the most frequent non-detected OmpA variant (OmpA-

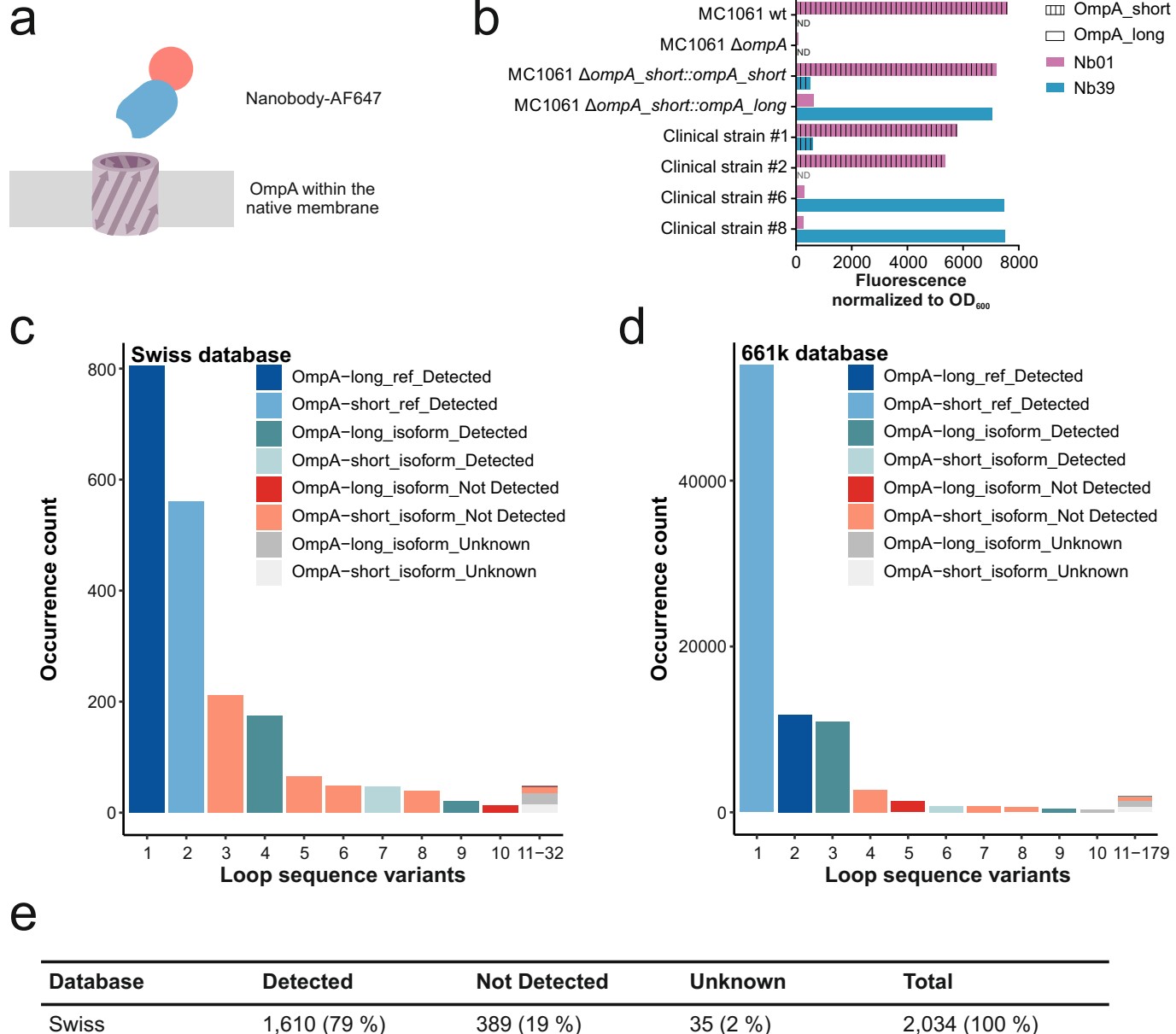

**Fig. 2 | Species coverage of OmpA nanobodies. a** Cellular binding assays were carried out using AlexaFluor647-labeled nanobodies. **b** Cellular binding assays using Nb01 and Nb39 (bars in violet or blue). The bar pattern indicates whether the respective strains express the OmpA-short or the OmpA-long isoform, respectively. ND: not determined. **c, d** Sequence analysis of extracellular loop regions of OmpA in the Swiss (**c**) or 661k (**d**) database. Unique OmpA sequences are listed according to their frequency found in the respective database. The reference OmpA-short and OmpA-long sequences were used to generated Nb01 and Nb39, respectively. Detected (blue/petrol), Not Detected (red), and Unknown detection (gray) indicates whether the respective OmpA variant is recognized by either Nb01 or Nb39. OmpA-long isoforms are colored in darker shades, and OmpA-short in lighter shades. **e** Summary of species coverage analysis.

ND#1), an OmpA-short isoform variant which accounts for 54% of the non-detected strains in the Swiss collection and 46% of the non-detected strains in the 661k dataset, features a NFDG motif in loop 3 instead of the more frequent NVYG motif. As the crystal structure revealed, the entire NVYG motif is recognized by Nb01 (Fig. 3c), thereby explaining why deviations in this core region of the OmpA-nanobody interface are not tolerated.

### Nb01 and Nb39 do not cross-react with other bacterial species

We performed a sequence analysis (see methods) to assess potential cross-reactions of our nanobodies by screening for protein sequences similar to each extracellular OmpA loop (i) within other *E. coli* proteins and (ii) within proteins of other bacteria. Within *E. coli*, no other matches were found. However, for other bacterial taxa, while no matches were found for loop 3, 46 matches were detected in OmpA homologs, with 2569 hits in *Enterobacteriaceae*, in 1174 hits in *Salmonella* and 852 hits *Klebsiella* strains of the loop 4 of OmpA (Supplementary Fig. 9).

To experimentally assess cross-reactivity of the nanobodies, we performed high-throughput flow cytometry measurements (Supplementary Fig. 7) to probe Nb01 and Nb39 against 19 reference strains belonging to species other than *E. coli*, of which 10 contain an annotated OmpA sequence. In addition, we tested our nanobodies against eleven reference

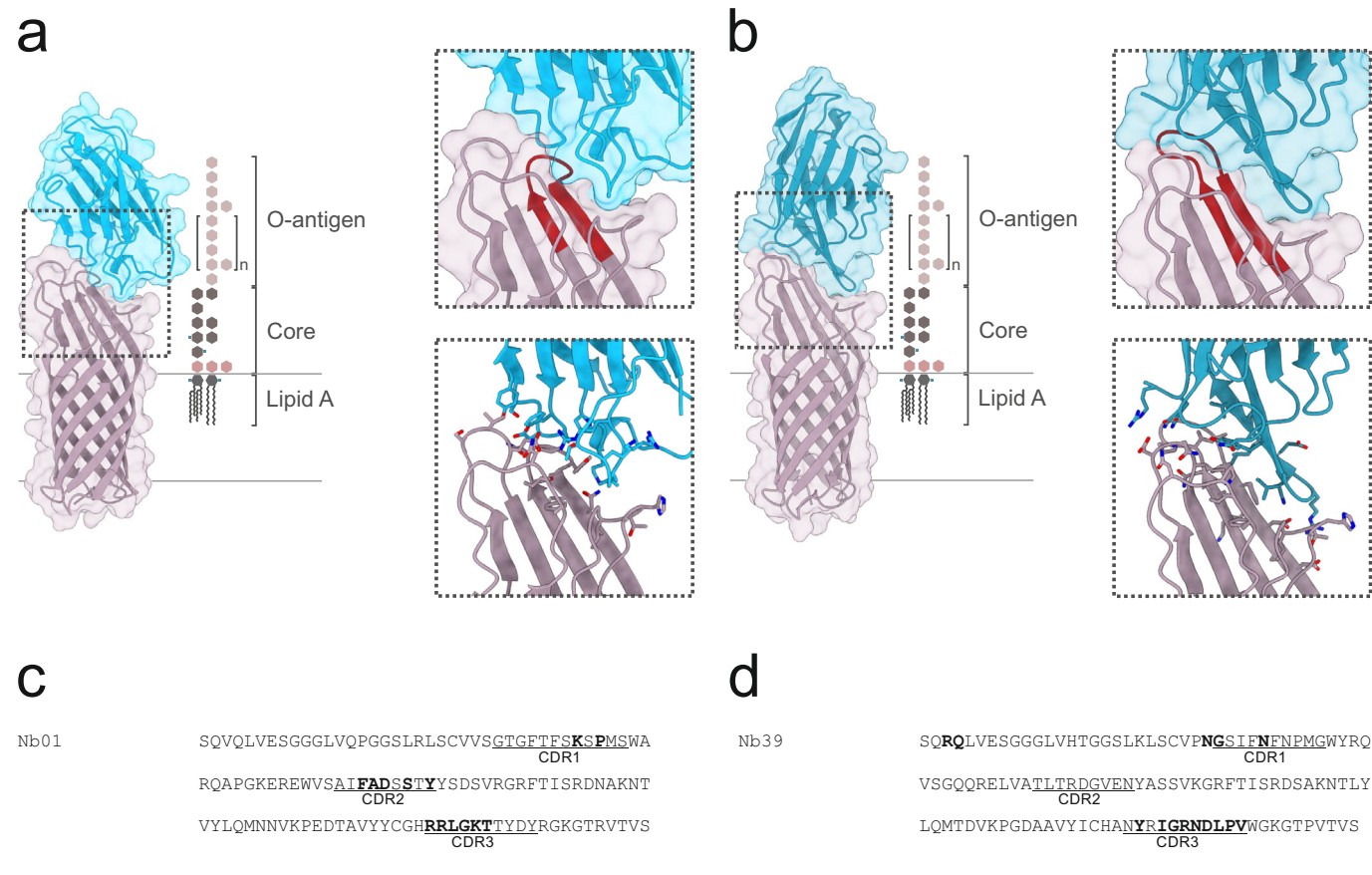

**Fig. 3 | Crystal structures of nanobody complexes. a, b** OmpA-short in complex with Nb01 (**a**) and OmpA-long in complex with with Nb39 (**b**) are shown as cartoons and transparent surfaces. OmpA is colored in purple, nanobody in blue. The dashed region is shown either with highlighted extracellular loop 3 (red) or with interfacing residues shown as sticks. LPS is shown as a reference to delineate the approximate lipid bilayer boundary and to indicate epitope shielding by sugar moieties. **c, d** Amino acid sequences of Nb01 and OmpA-short (**c**) and Nb39 and OmpA-long (**d**) with interacting residues highlighted in bold. The three CDRs of the nanobodies and the four extracellular loops of the OmpA are underlined and labeled.

strains for *E. coli* as well as three *Shigella* strains (which are in fact specialized diarrheagenic *E. coli*) (Supplementary Table 8). Our nanobodies demonstrated to reliably bind all *E. coli* reference strains and the three *Shigella* strains. Of further note, the OmpA-short and OmpA-long isoforms are distributed quite evenly over the tested reference strain set, showing that nanobodies against both isoforms are needed to reach satisfactory species coverage. No cross-reactive binding to any of the 23 tested non-*E. coli* strains was observed. This finding correlates well with the bioinformatics analysis (Supplementary Fig. 9), because the respective OmpA sequences differed in loop 3 from the two *E. coli* OmpA reference sequences.

**Nanobodies bind their target protein with high affinity**
To characterize binding of Nb01 and Nb39 in the cellular context of the *E. coli* lab strain and clinical isolates, we employed flow cytometry using fluorescently labeled nanobodies (Fig. 4, Supplementary Fig. 10a). Both nanobodies showed bright OmpA-specific staining within the native bacterial cell environment, even in the presence of O-antigens found on clinical strains (Fig. 4a and b). Notably, both nanobodies exhibited high specificity for the targeted OmpA isoform, i.e. Nb01 only binds to strains expressing OmpA-short (CS#1 and CS#2), but not to strains expressing OmpA-long (CS#6 and CS#11) and vice versa for Nb39. To obtain values for the apparent binding affinity of Nb01, we determined EC50 values, i.e. the nanobody concentration at which the half-maximal binding signal in the flow cytometry analysis is reached. For the *E. coli* lab strain MC1061, an EC50 of 12.95 ± 3.24 nM was determined (Fig. 4c). In contrast, the EC50 values were around 2–5 times weaker for clinical isolates CS#1 and CS#2 (25.89 nM ± 8.54 nM and 61.10 nM ± 28.71 nM, respectively). Since the extracellular loops of OmpA-short of CS#1 and CS#2 do not differ from the reference OmpA-short sequence present in *E. coli* MC1061 (Supplementary Fig. 3), the weaker EC50 values can be attributed to the presence of the O-antigen layer in these strains. To test this hypothesis, we pre-treated CS#2 with EDTA, which destabilizes the LPS by chelating divalent cations[32] and which is also used as anticoagulant in blood samples. Indeed, supplementation with EDTA during staining resulted in a ten-fold enhancement of the EC50 values for Nb01 targeting CS#2 (4.98 ± 0.48 nM; Fig. 4d). Additionally,

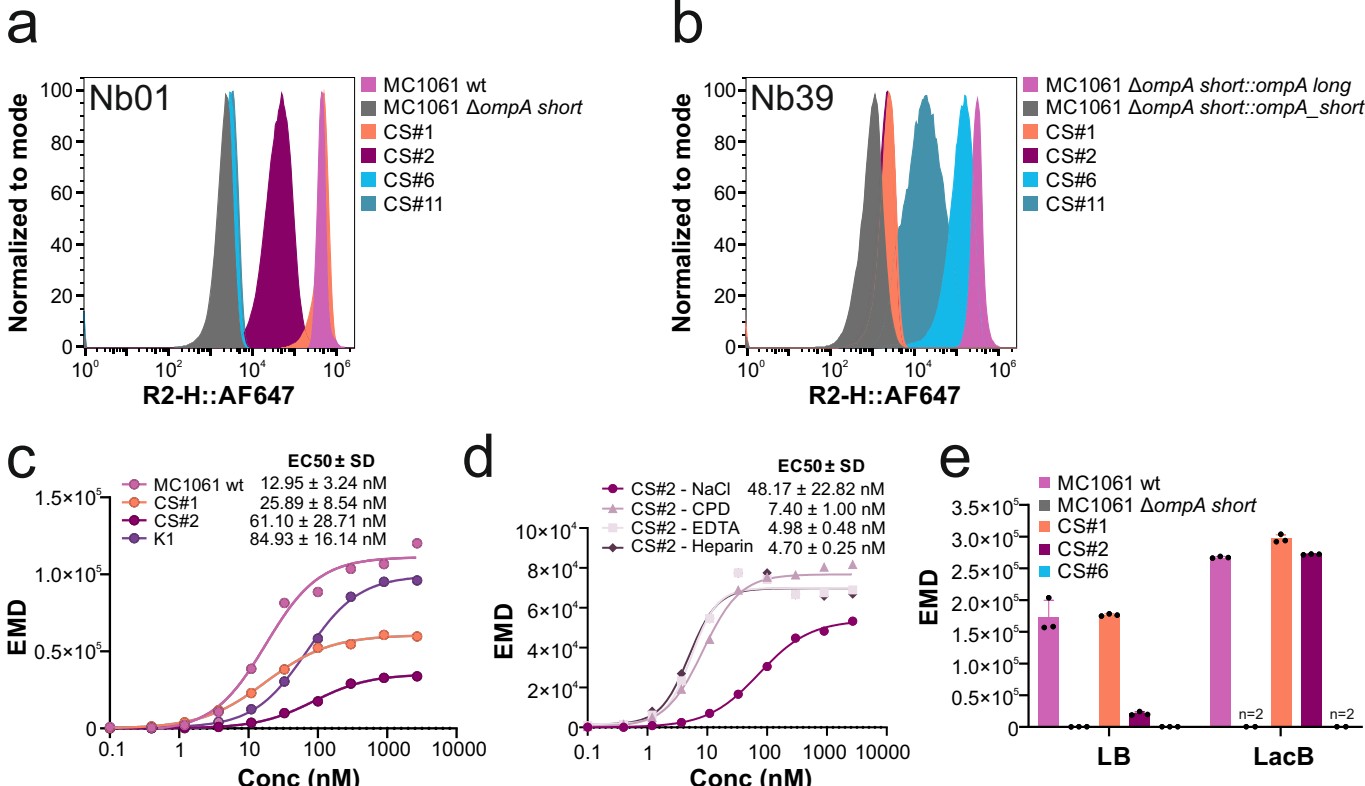

**Fig. 4 | Nanobody characterization using flow cytometry. a, b** Flow cytometry histograms showing binding of Nb01-AF647 (**a**) or Nb39-AF647 (**b**) to the indicated bacterial strains. For these experiments, 33 nM of fluorescently labeled nanobodies were used. Measurements were performed on an Aurora flow cytometer (Cytek) using unfixed cells. **c, d** Concentration-dependent binding of Nb01-AF647 against the indicated strains, including the *E. coli* K1 ATCC 11775 strain producing a K1 polysaccharide capsule (**c**) or against CS#2 in the presence of NaCl or different anticoagulants (CPD, EDTA, or Heparin) (**d**). Data were fitted with the Hill equation to determine EC50 values. Cells were fixed and samples measured in triplicates on the CytoFlex S flow cytometer (Beckman). **e** Binding of Nb01-AF647 (10 nM) to the indicated strains grown in Luria Broth (LB) or Lactose Broth (LacB) media. Measurements were performed in triplicates on an Aurora flow cytometer (Cytek) using unfixed cells. For all data shown, earth mover's distances (EMDs) were calculated relative to binding of an unrelated nanobody added at 10 nM (see methods). Representative data of three biological replicates are shown. Errors correspond to standard deviations of three biological replicates, which correspond to three independently inoculated cultures grown and processed in parallel. CS clinical strain, wt wildtype, EC50, half maximal effective concentration.

we tested the effects of the anticoagulants citrate phosphate dextrose (CPD) and heparin, which improved EC50 values for Nb01 on CS#2 as well (7.40 ± 1.00 nM and 4.70 ± 0.25 nM, respectively; Fig. 4d). Finally, we asked whether nanobodies can cross the capsular layer. To this end, we performed FACS staining experiments using the well-studied *E. coli* K1 strain ATCC 11775, which has been shown to produce a dense K1 polysaccharide capsule[33–35]. The periplasmic loop regions of *E. coli* K1 OmpA-short (Supplementary Table 1) are identical to the ones of *E. coli* MC1061 OmpA-short against which we generated Nb01. We determined an EC50 of 84.93 ± 16.14 nM for Nb01 binding to *E. coli* K1 (Fig. 4c), which is in a similar range as observed for CS#2 and clearly shows that nanobodies can cross the capsular layer. Notably, the maximal signal intensity at high nanobody concentrations indicates good accessibility of OmpA in *E. coli* K1. We wish to note that we did not have an isogenic strain pair available to directly assess the contribution of the capsule to nanobody binding. Further, we did not assess whether the clinical strains we analyzed in the context of this study possessed capsules. Hence, the relative contribution of the capsule versus the O-antigen layer to interfere with nanobody binding to OmpA were not addressed in this study.

## LPS density depends on growth medium

By serendipity, we discovered that when clinical *E. coli* strains were grown in Lactose Broth (LacB) instead of Luria Broth (LB), staining with AF647-labeled nanobody is drastically enhanced, particularly for CS#2 (Fig. 4e). We

suspected that the change of growth medium alters the LPS structure. To validate this hypothesis, we extracted the LPS from three clinical strains grown either in LacB or LB, separated them on an SDS-PAGE gel, and stained the O-antigen sugars with Emerald dye (Fig. 5a). We used the lab strain *E. coli* MG1655 as a reference strain devoid of O-antigens. As positive control, we included the *E. coli* MG1655 *wbbL*+ strain, whose O-antigen production was genetically restored[36,37]. When grown in LB, we noted a large variability among the clinical isolates in terms of the amount of O-antigen produced, with CS#1 clearly having the least O-antigen presented on the cellular surface. When grown in LacB, the O-antigen density was strongly reduced for *E. coli* MG1655 *wbbL*+ and CS#1 and CS#2, and to a lesser degree also for CS#6. In addition, a prominent O-antigen ladder was observed for CS#1 and CS#2. To further assess how LacB influences O-antigen surface density, we grew CS#2 in LB medium or blood culture medium supplemented with the respective components of LacB, namely beef extract, lactose and peptone. We performed flow cytometry measurements to detect OmpA accessibility under conditions where staining of cells grown in LacB versus LB is strongly enhanced (Fig. 5b). Thereby, we found that supplementation of lactose to either LB or blood culture medium greatly enhances OmpA accessibility, as shown by increased flow cytometry signal intensity (Fig. 5b). In summary, we noted that the extent of O-antigen surface decoration is highly strain-dependent, and that the growth conditions and in particular supplementation of lactose to the growth medium have a profound effect on the O-antigen density and length distribution and thus, nanobody binding.

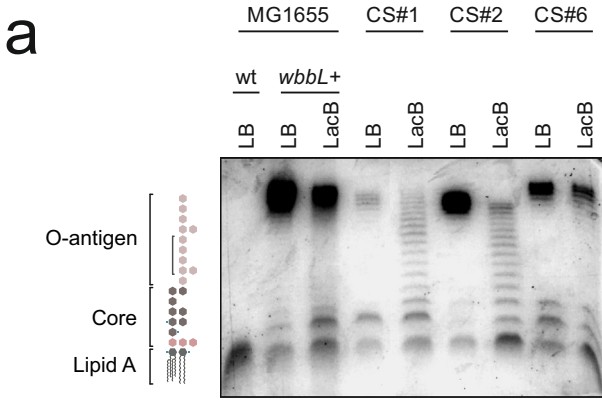

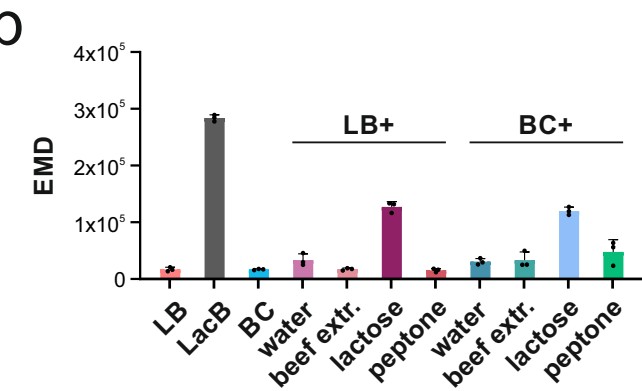

**Fig. 5 | Analysis of LPS and OmpA accessibility in clinical strains. a** LPS analysis of *E. coli* lab strain MG1655 and its isogenic strain with restored O-antigen, MG1655 *wbbL +*, as well as from the indicated clinical strains cultured in either LB or LacB. Extracted LPS was separated using SDS-PAGE and its sugars were detected using Emerald staining. **b** Binding of Nb01-AF647 (10 nM) to CS#2 grown in Luria Broth (LB), Lactose Broth (LacB), blood culture (BC) media, or LB or BC medium supplemented with any of the components of LacB medium. Measurements were performed on an Aurora flow cytometer (Cytek) using unfixed cells. Earth mover's distances (EMDs) were calculated relative to binding of an unrelated nanobody added at 10 nM (see methods). Data points correspond to three independently inoculated cultures grown and processed in parallel. Error bars correspond to standard deviations. CS clinical strain, wt wildtype.

## Nanobody-mediated capture of *E. coli* lab strain

To establish a bacterial capture assay at a bacterial load that is realistic in patients suffering from BSI, we first diluted the lab strain *E. coli* MC1061 to a cell density of ~500 CFU/mL in PBS. For magnetic extraction, we initially immobilized nanobodies via a short maleimide-PEG$_{11}$-biotin linker (Fig. 6b) on magnetic streptavidin beads and then used these functionalized beads to capture bacteria (Fig. 6a, procedure 1). Thereby, we achieved complete capture of the *E. coli* MC1061 lab strain within 2 h, while the corresponding Δ*ompA* strain was not captured (Fig. 6c). Complete capture was also achieved upon lowering the bacterial cell number to 50 CFU/mL (Fig. 6d). The large data variation is attributed to the small number of cells involved in this assay. Disappointingly, we failed to capture any of the clinical strains under the same experimental conditions.

## Targeting trimeric OmpF with nanobodies

With the aim to overcome target accessibility problems encountered when capturing clinical strains decorated with a dense O-antigen layer, we generated nanobodies against the outer membrane protein OmpF. While OmpA exists as monomer in the outer membrane, crystal structures of OmpF[38] as well as biochemical experiments demonstrated its homotrimeric assembly[39]. In addition, the OmpF monomer is a β-barrel consisting of sixteen β-sheets, as opposed to eight β-sheets in case of OmpA. According to a recent molecular simulation study, the footprint of OmpD (which akin to OmpF is trimeric) displacing LPS within the outer membrane is around six times larger than the one for OmpA in *Samonella spp.*[40], a bacterial species that is closely related to *E. coli*. Using our nanobody generation pipeline, we targeted purified *E. coli* OmpF, and identified around 10 strong binder candidates as based on flycode analysis (Supplementary Tables 6, 9 and 10). One of the OmpF nanobodies, called Nb18, was further characterized using flow cytometry, thereby determining an EC50 of 3 nM against the *E. coli* K12 strain (Supplementary Fig. 11c). However, maximal binding intensities of nanobodies targeting OmpF were 10 times lower than for nanobodies targeting OmpA, which can be explained by the lower expression level of OmpF (Supplementary Fig. 11d). No binding towards *E. coli* MC1061 Δ*ompF* was detected (Supplementary Fig. 11a and b).

In attempts to capture *E. coli* MC1061 with Nb18 immobilized via a short maleimide-PEG$_{11}$-biotin linker on magnetic streptavidin beads, capture was only partial (Supplementary Fig. 11e). This stands in contrast to the near 100% capture using Nb01 directed against OmpA under identical experimental conditions (Fig. 6c), indicating that higher OMP densities are beneficial to achieve efficient capture and that the encounter of functionalized beads with the bacterial cell is likely the rate-limiting factor. It was therefore not surprising that capture of clinical strain CS#1 via Nb18-functonalized beads was not successful either (Supplementary Fig. 11e).

## Linker engineering to bridge the LPS

Since our fluorescently labeled nanobodies efficiently bind to O-antigen containing *E. coli* strains (Fig. 4), we reasoned that steric hindrance caused by the LPS layer might impede the capture of clinical *E. coli* strains. To test this hypothesis, we generated Avi-tagged nanobody constructs with the Avi-tag being separated with linkers of variable lengths and flexibility from the nanobody core (Fig. 6b). The linkers included flexible glycine and serine linkers (GS-linkers)[41], helical structures, containing the leucine-glutamate-alanine sequence (LEA-linker)[42,43], and stiff linkers containing proline-alanine-proline-alanine repeats (PAPA linker)[43,44]. As one turn with 3.6 amino acids in an α-helix covers a distance of 0.54 nm, the LEA-linker, with its 46 residues, covers a distance of ~7 nm. Assuming fully extended peptides, adjacent residues are 0.35 nm apart. Therefore, the PAPA-linker with 33 residues is maximally 11.5 nm long. A fully extended GS-linker would cover 7 nm, but its flexible nature likely makes it shorter.

Avi-tagged nanobodies were then purified and enzymatically biotinylated to be recognized by Atto565-labeled streptavidin. With a dimension of around $4.5 \times 5 \times 5$ nm$^3$, the streptavidin tetramer is much larger than the AF647 dye and prevents the nanobodies (having themselves a dimension of around $2 \times 2 \times 4$ nm$^3$) from binding to OmpA in clinical strains due to steric hindrance as opposed to nanobodies labeled with AF647 (Supplementary Fig. 5). Hence, size enlargement via streptavidin binding impedes the penetration of nanobodies through the O-antigen layer to reach their target (Supplementary Fig. 6). For our binding assays, we included the lab strain MC1061 (devoid of O-antigen), CS#1 featuring a loose and partially permeable O-antigen layer and CS#2 being shielded with a dense and thus challenging O-antigen layer (Fig. 5a). Lab strain MC1061 lacking the respective OMPs served as negative control in all assays performed. In a first experiment, constructs having Nb01 as binding module extended with the short PEG$_{11}$-biotin linker or the longer polypeptide linkers GS, LEA or PAPA were complexed with Atto565-labeled streptavidin and tested (Supplementary Fig. 12a). As expected, for the lab strain MC1061, equally strong binding signals were observed for all linker designs, whereas the signal was absent when probing the MC1061 Δ*ompA* strain. In case of CS#1, the construct with the short PEG$_{11}$-biotin linker gave rise to the maximal binding signal, showing that in this clinical strain steric hindrance does not appear to be an issue. In contrast, the construct with the short PEG$_{11}$-biotin linker did not give rise to any binding signal in case of CS#2,

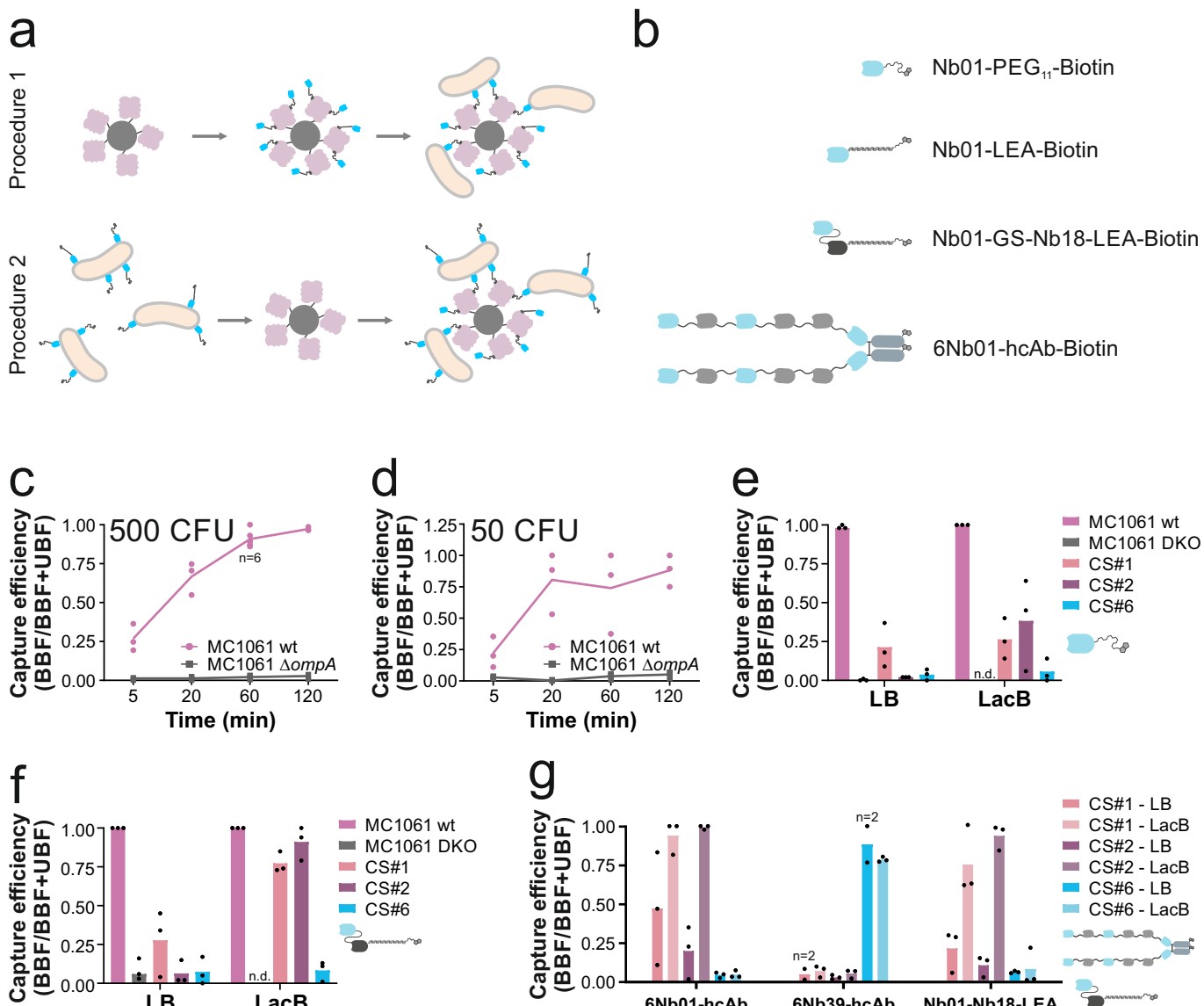

**Fig. 6 | Immunomagnetic capture of *E. coli*. a** Experimental scheme of capture assay. In procedure 1, magnetic streptavidin beads are saturated with biotinylated nanobodies in the first step. Next, they are added to a solution with bacteria for capture. In procedure 2, bacteria are first incubated with biotinylated nanobodies and in a second step, magnetic streptavidin beads capture the tagged bacteria. **b** Schematic illustration of nanobody constructs. Nb01 is colored light blue, Nb18 black and other nanobodies in gray. Biotin is depicted as chemical structure. PEG11 is shown as zagged line. Stiff linkers are shown as α-helices, flexible glycine-serine linkers as lines. Fc-parts of antibodies are shown as gray, rounded rectangles. 6Nb01-hcAb is comprised of Nb01-Nb02-Nb01-Nb10-Nb02-Nb01-Fc-fragment. **c, d** Time-dependent capture of lab strain *E. coli* MC1061 according to procedure 1 using Nb01-PEG11-biotin. Error bars correspond to standard deviations of

biological triplicates (except the indicated data point where $n = 6$). **e** Capture of *E. coli* cells (ca. 500 CFU) cultured in LB or LacB using Nb01-PEG11-biotin (see icon). Capture was performed for 1.5 h according to procedure 2. **f** Capture of *E. coli* cells (ca. 500 CFU) cultured in LB or LacB using Nb01-Nb18-LEA-biotin (see icon). Capture was performed for 1.5 h according to procedure 2. **g** Capture of *E. coli* cells (ca. 500 CFU) cultured in LB or LacB using 6Nb01-hcAb-biotin (see icon) or 6Nb39-hcAb-biotin, as indicated. Capture was performed for 1 h according to procedure 2. EDTA (0.5 mM) was added to capture buffers. n.d., not determined. **e−g** Data points correspond to results of capture experiments performed on different days ($n = 3$, unless indicated otherwise). Capture efficiencies were determined by dividing the bead-bound fraction (BBF) by the sum of the BBF and unbound fraction (UBF) counted in the supernatant.

indicating a strong shielding effect of the dense LPS. Interestingly, while the construct with a flexible GS linker did not reach OmpA in CS#2, the constructs bearing the more rigid LEA and PAPA linkers were partially able to penetrate and bridge the dense LPS of CS#2 (Supplementary Fig. 12a). The same experiment was also conducted with constructs wherein the binding module was Nb18, hence targeting OmpF. Quite unexpectedly, we observed only very weak binding for CS#1 as well as CS#2 as compared to the lab strain MC1061, independently of the linker used (Supplementary Fig. 12b). Finally, we generated a construct series wherein Nb01 is fused to Nb18 with a flexible glycine linker, followed by

either LEA, PAPA or GS linkers and a biotinylated Avi-tag (Fig. 6b). The performance of this construct in the cellular binding assay with Atto565-labeled streptavidin was similar to the construct bearing only Nb01 as binding module (Supplementary Fig. 12c), showing that the increased length gained by the additional nanobody in the construct does not further improve cellular staining in a discernible manner.

Based on our findings with the nanobodies we have analyzed as part of this study, we concluded that (i) OmpA is better suited as a target to stain and capture *E. coli* than OmpF, (ii) structured and stiff linkers are potentially suited to bridge the LPS layer of clinical *E. coli* strains featuring dense

O-antigen decoration as found in CS#2 and (iii) the linker length is an important parameter if one attempts to capture clinical strains.

### Capture of clinical *E. coli* strains using engineered heavy chain-only antibodies

Based on our systematic assessment of linker designs, we concluded that achieving an even greater separation between the nanobody and the magnetic bead would likely be required for a reliable capture of clinical strains. Therefore, we constructed heavy chain-only antibodies (hcAbs), wherein we fused up to six OmpA-nanobodies as "pearls on a string", starting with Nb01 or Nb39 as the outermost molecule (Fig. 6b).

Further, we reasoned that the experimental setup how the binding assay is performed could potentially influence capture efficiency (Fig. 6a). Therefore, we compared bacterial capture with magnetic beads functionalized with nanobody constructs (procedure 1 in Fig. 6a) with a procedure wherein the biotinylated nanobody construct is added in solution and allowed to bind the cells, followed by capture using streptavidin-coated magnetic beads (procedure 2 in Fig. 6a). When assessing the two procedures side-by-side with a hcAb, procedure 2 proved to be more efficient (Supplementary Fig. 13a). Further, we showed that the concentration of the hcAb needs to exceed the estimated concentration of OmpA by at least 10-fold to achieve successful capture according to procedure 2 (Supplementary Fig. 13b).

To systematically assess the potential benefits of LPS trimming and thinning, we grew the cells in LB or LacB, and first captured the cells using Nb01 modified with the small $PEG_{11}$-linker according to procedure 2. Thereby, we found that in particular for CS#2, growth in LacB leads to improved capture (Fig. 6e). Next, we performed the same assay with the Nb01-Nb18-LEA-biotin construct, which is considerably longer than Nb01-$PEG_{11}$-biotin. Intriguingly, in this case we saw further improved capture of CS#1 and CS#2 when the cells were grown in LacB, as compared to those grown in LB (Fig. 6f), clearly showing that LPS trimming and linker extension are both required to achieve successful capture. To further increase linker length, we used two hcAb constructs with six OmpA-nanobodies separated by alternating the helical LEA-linker and the rigid PAPA-linker, and with the outermost nanobody being either Nb01 or Nb39 to either target OmpA-short or OmpA-long, respectively (Fig. 6b, complete sequences in Supplementary Fig. 14). Again, capture of cells grown either in LB or LacB was assessed, revealing for the first time successful and reliable capture of clinical strains CS#1, CS#2 and CS#6 grown in LB (Fig. 6g). Capture of cells grown in LacB by the hcAb constructs was consistently high.

In summary, our data show that capture of clinical *E. coli* strains is increasingly feasible if one generates nanobody constructs with long spacer lengths, systematically optimizes the capture procedure or chooses growth media that reduce LPS decoration.

## Discussion

Molecular tools for the reliable and rapid detection, enrichment and isolation of live *E. coli* cells offer broad applications in routine diagnostics, surveillance and research. However, the high sequence diversity within the versatile *E. coli* species have thus far prevented the development of antibodies or other capture molecules, that would reach satisfactory species coverage.

In this study, we took advantage of the small nanobody scaffold to target the conserved and abundant outer membrane protein OmpA on the cellular surface of *E. coli*. In clinical isolates, access to OmpA is shielded by the dense O-antigen sugars, as we demonstrated in experiments in which we attached the nanobodies to larger moieties such as streptavidin. Hence, binding of classical antibodies to OmpA is at least partially impeded, and explains the small number of biomolecules thus far generated against *E. coli* OmpA that would reliably work in the context of intact cells.

Only few nanobodies targeting surface structures on pathogenic bacteria have been described in the literature[22,45–50]. In previous studies, immunizations were performed using heat-inactivated *Campylobacter*

species[45] or fixed *Acinetobacter baumannii* cells[47]. While in case of *Campylobacter*, the identified nanobodies were shown to recognize the major outer membrane protein (MOMP)[45], the target structure remained unidentified in case of *Acinetobacter baumannii*[47]. In both studies, cellular staining with the nanobodies required cell fixation, indicating that the targeted epitopes are not accessible in the context of the living cell. In case of the *A. baumannii* nanobodies, the authors demonstrated that the peptide corresponding to CDR3 can reach the target epitope in the context of intact cells.

Our approach taken here and in a previous work[22] was to first raise nanobodies against a purified OMP, thereby achieving strong enrichment of the nanobody pool against the specific target. In a subsequent step, we employed the flycode technology[22] to deep-screen the nanobody pool against the respective OMP in intact cells, thereby identifying nanobodies that perform exceptionally well in the cellular context of a broad set of strains of the same species. Our approach comes with the important advantage that we knew the target structure prior to the selection. Hence, using bioinformatic analyses we could rationally plan the selection campaign to reach high species coverage with a low number of nanobody binders. As we demonstrated in this study, our targeted approach allowed us to reach a species coverage of *E. coli* of above 90% using only two nanobodies.

In addition, our bioinformatic analysis allowed us to identify the expected gaps we would have to fill to further improve species coverage. Firstly, we realized that the *ompA* gene appears to be lacking in around 5% of the genomes found in the 661k database. The reasons behind missing *ompA* sequences might be manifold, including general errors in the sequencing and assembly of genomes. Nanobody binding experiments on strains apparently lacking OmpA would be needed to confirm this in silico finding. The lack of the *ompA* gene in clinical isolates is puzzling, because OmpA has been described to be an important immune evasin[51]. Should the proportion of *E. coli* strains without *ompA* indeed amount to around 5% as estimated, the nanobodies raised against OmpF, such as Nb18, could be used to fill the gap.

Among the *E. coli* strains in which we found an *ompA* gene, our two nanobodies reach a coverage of 91% in the 661 k database. As a comparison, the current standard method to detect *E. coli* in environmental samples based on β-D-glucuronidase activity detects 94% of *E. coli* and 42% of *Shigella* strains[52]. By analyzing the OmpA sequences that escaped detection by Nb01 and Nb39, we anticipate that another 3 – 4 nanobodies are required to achieve a coverage of close to 100% of the strains producing OmpA. To the best of our knowledge, no other studies exist wherein species coverage of antibodies or nanobodies raised against bacteria have been systematically analyzed and tested to the extent described in this work.

The identified nanobodies were thoroughly characterized at the biochemical and molecular level to rationalize their binding mechanism in the cellular context. The crystal structures of Nb01 and Nb39 revealed that they recognize all four extracellular loops of the respective OmpA isoform, thereby explaining their specificity for OmpA-short and OmpA-long, respectively. Intriguingly, in both structures the nanobodies extend the OmpA barrel like a "flame on a candle", and hence avoid strong steric clashes with the LPS sugars. Flow cytometry analysis confirmed nanobody binding in the context of clinical isolates and showed that their apparent binding affinities are in the single to double digit nanomolar range, depending on the strain used.

While the nanobodies performed very well for cellular staining, immunomagnetic capture of clinical strains turned out to be more challenging, indicating that the O-antigen layer and/or the capsule form steric barriers. In a series of systematic experiments, we could show that both the O-antigen barrier as well as the capsular barrier permit the passage of nanobodies but obstructs bulkier molecules like streptavidin and even larger magnetic beads from penetrating effectively. To enable immunomagnetic capture of clinical strains, we extended the distance between the nanobodies and the bulky magnetic beads by engineering nanobody constructs with long linkers. Accordingly, we achieved the most successful capture results with our longest hcAb constructs. In addition, we discovered that *E. coli* cells

cultured in lactose broth produce a much thinner sugar layer and are much easier to capture. Lactose broth consists of beef extract, lactose and peptone and we were able to show that the addition of lactose to growth medium exerts the effect of LPS thinning through a molecular mechanism that remains elusive. This finding could be highly relevant for the implementation of bacterial capture steps of early blood cultures before they turn positive.

Our nanobodies hold promise to be used for diagnostic applications. The enrichment of live bacteria allows for seamless integration with phenotypic antimicrobial susceptibility testing (AST) using e.g. microfluidics, expediting diagnostics by eliminating or shortening blood culture. In molecular diagnostics, the identified nanobodies can be used to enrich bacteria directly from patient samples, thereby improving the ratio of bacterial DNA over human DNA and rendering next generation sequencing based techniques more efficient. Further, our nanobodies are expected to perform well for immunostaining in the context of diagnostic digital microscopy. They may also find application as *E. coli*-specific staining agents upon total bacterial capture using the synthetic beta-2-glycoprotein I peptide, which recognizes a broad range of microbial pathogens[53]. Beyond clinical diagnostics, the described *E. coli* nanobodies are highly suited for culture-independent water and food surveillance. Finally, we are confident that our nanobodies will find broad application in clinical research as a highly sensitive tool to live-stain clinical *E. coli* isolates in the context of infected cells, tissues or organs-on-a-chip.

## Material and Methods
### Strains used in this study
In this study we used *E. coli* K-12 strain MC1061, for which we generated unmarked gene deletions as outlined below. *E. coli* K-12 strain MG1655 wt and *wbbL+* were kindly provided by Prof. Dr. Sebastian Hiller. Clinical *E. coli* strains used in this study are whole genome sequenced clinical isolates collected at the Institute of Medical Microbiology, University of Zurich.

### Generation of gene deletions
To generate gene deletions in *E. coli* MC1061, (NCBI:txid1211845) a two-step recombination technique was used as described[54–56]. Firstly, the target gene was replaced by homologous recombination with a FRT-flanked kanamycin cassette using the red recombinase from the pKM208 plasmid[54,55]. Secondly, the kanamycin cassette was removed by expressing a flippase from the pSIJ8 plasmid[56]. To allow for homologous recombination, the kanamycin resistance gene in between two FRT sites was flanked with homologous DNA stretches upstream and downstream of the *ompA* or *ompF*. To generate this kanamycin cassette, we generated three PCR products which we ligated together and integrated into a pINIT vector containing an Chloraphenical resistance (Addgene: #46858) using FX cloning[57] resulting in pINIT_KO_ompA and pINIT_KO_ompF. The three PCR products composed of the 5' region upstream (primer pairs of ompA_EC_FX_5_FW/RV and ompF_EC_FX_5_FW/RV), the Kanamycin cassette (primer pair Kan_cassette_FW/RV), and the 3' region downstream (primer pairs of ompA_EC_FX_3_FW/RV and ompF_EC_FX_3_FW/RV), of the region of interest. Finally, PCRs of the generated pINIT_KO_ompA and pINIT_KO_ompF were carried out, resulting in double stranded PCR products encoding for upstream homologous region-FRT-Kan$^R$-FRT-downstream homologous region (PCR primers listed in Supplementary Table 11) using primer pairs ompA_EC_DKO_5_FW and ompA_EC_DKO_3_RV, and ompF_EC_DKO_5_FW and ompF_EC_DKO_3_RV. *E. coli* MC1061 cells were transformed with pKM208 (carrying an ampicillin resistance marker) and grown to $OD_{600\,nm} = 0.1$ at 30 °C and 160 rpm, at which point recombinase expression was induced with 1 mM IPTG. After 4 h expression, cells were made electrocompetent by washing three times with ice-cold water. Subsequently, electroporation was performed using 3 ug of the gene deletion PCR product, followed by recovery in LB and selection on LB/Kan[50] plates. Kanamycin resistant clones were selected, and integration confirmed by colony PCR. To remove the heat-sensitive pKM208 plasmid, the selected colonies were cultured at 42 °C. In the second step, Kanamycin-resistant but Ampicillin-sensitive clones were made electrocompetent and transformed with pSIJ8 (carrying an ampicillin resistance marker). Again, cells were grown to $OD_{600\,nm} = 0.1$ at 30 °C and 160 rpm, and flippase expression was induced by adding 10 mM Rhamnose. After 6 h of expression at 30 °C, the cells were plated on LB Agar plates. Clones were picked and grown at 42 °C and 160 rpm for 4 h to lose the heat-sensitive pSIJ8 plasmid and plated again on LB Agar plates. Gene deletions of Kanamycin- and Ampicillin-sensitive clones were confirmed by Sanger sequencing.

### Protein expression, purification and biotinylation
Genes encoding *ompA-short* and *ompF* were amplified from *E. coli* K-12 MC1061, while *OmpA-long* was amplified from clinical isolate #11 (CS#11) of IMM (Supplementary Table 1). Primer sequences are shown in Supplementary Table 11. Primer pair ompA_FW/ompA_RV was used for amplifying full length *ompA-short* and *ompA-long*, primer pair ompA_FW/ompA_TMD_RV for transmembrane domain of *ompA-short* and *ompA-long*, and primer pair ompF_FW/ompF_RV for full length *ompF*. For OmpF, a point mutation (S75C) allowing for site-specific biotinylation was introduced (OmpF_S75C_FW &RV primers).

Full length OmpA-short and OmpA-long were expressed in p7XC3H (Addgene: #47065, containing a C-terminal 3C-cleavage site and deca-His-tag) for immunization or p7XCA3H (like p7XC3H but with an Avi-tag before the 3 C cleavage site) for phage display selection. Transmembrane domain of OmpA-short and OmpA-long for X-ray crystallography and full-length OmpF_S75C for immunization and selection were expressed in p7X (tagless). All constructs were expressed as inclusion bodies in *E. coli* C43 cells.

Cells were grown in TB/Kan[50] to $OD_{600\,nm}$ of 0.9–1.5 at 37 °C and 90 rpm in baffled flasks. Expression was induced with 1 mM IPTG, followed by an additional 3 h growth before harvesting by centrifugation (6000 × g, 10 min, 4 °C). Pellets were resuspended in 20 mM Tris, pH 8 and cells were disrupted by passing 4 times through a microfluidizer at 30 kpsi. Unbroken cells were removed by low centrifugation (2000 × g, 15 min, 4 °C). Inclusion bodies were harvested (8000 × g, 10 min, 4 °C), washed once with 20 mM Tris, pH 8, 1% Triton X-100 and twice with 20 mM Tris, pH 8. Inclusion bodies were solubilized in 20 mM Tris, pH 8 and 6 M guanidine hydrochloride for 2 h at room temperature. For full-length and transmembrane domain OmpA (short and long isoforms), aggregates were spun down (8000 × g, 30 min, 4 °C), and the supernatant was added dropwise to 20 mM Tris, pH 8, 5% $C_8POE$ to finally reach a 1:12 dilution, and incubated for 2 h at room temperature for folding. For full-length OmpA, refolded proteins were subjected to Ni-NTA column, washed with 15 column volumes (CV) 20 mM Tris, pH 8, 0.5% $C_8E_4$, 50 mM imidazole, and eluted with 4 CV 20 mM Tris, pH 8, 0.5% $C_8E_4$, 250 mM imidazole. 3 C protease was added and His-tag was cleaved over night at room temperature while dialyzing against 20 mM Tris, pH 8, 0.5% $C_8E$. Cleaved protein was further purified via reverse IMAC, concentrated, and applied to size-exclusion chromatography (SEC) using a Superdex S200 increase 10/300 GL column in TBS, pH 7.4, 0.5% $C_8E_4$. In case of the tag-less transmembrane constructs of OmpA expressed from p7X, refolded proteins were concentrated, and supernatants were directly run on SEC.

The Avi-tagged versions of OmpA-short and OmpA-long were biotinylated in vitro using purified BirA protein[58]. The biotinylation reaction was carried out after reverse IMAC in TBS pH7.5, 0.5% $C_8E_4$, 5 mM ATP, 10 mM MgOAc and two-fold molar excess of biotin overnight at 4 °C, followed by SEC.

For OmpF_S75C, solubilized inclusion bodies were applied to a PD10 column and eluted with 50 mM Tris, pH 8, 6 M Urea. The protein was diluted to 15 mM Tris, pH 8, 6 M Urea and subjected to anion exchange chromatography using a Resource™ Q, 1 mL column and eluted via a gradient from 15 mM bis-Tris, pH 7, 6 M Urea to 1 M NaCl, 15 mM bis-Tris, pH 7, 6 M Urea. Fractions corresponding to (unfolded) OmpF_S75C were combined and added dropwise to 50 mM Tris, pH 8, 1 mM DTT, 0.1 M EDTA, 0.2% β-DDM (1:20 dilution) and incubated at

37 °C overnight. The protein was then concentrated and applied for SEC using a Superdex S200 increase 10/300 GL column in TBS, pH 7.5, 0.05% β-DDM.

For biotinylation of OmpF_S75C, following refolding, the protein was concentrated, and buffer was exchanged to TBS, pH 7.5, 0.1% β-DDM using PD10 columns. A five-fold molar excess of biotin-maleimide (Sigma, B1267) was added, and the reaction proceeded for 1 h at room temperature before concentration and application of SEC using a Superdex S200 increase 10/300 GL with TBS, pH 7.5, 0.05% β-DDM.

For the expression of OmpA in the outer membrane for cellular binding assays or flow cytometry experiments, *E. coli* MC1061 Δ*ompA* were transformed with plasmids pBXNPH3 containing either *ompA-short* or *ompA-long*. For expression, cells were grown in TB/Amp[120] at 37 °C and 90 rpm in baffled flasks to an $OD_{600\,nm}$ of about 0.5. The temperature was lowered to 20 °C and cells were grown for an additional 2 h. Expression was induced with 0.05% L-arabinose at an $OD_{600nm}$ of 0.8-1.2. Expression was carried out overnight.

### Alpaca immunizations
Alpacas were immunized by a total of four subcutaneous injections of about 100 ug purified OmpA-short, OmpA-long or OmpF at 2-week intervals. Small blood samples were taken to follow the immune response by ELISA (Supplementary Fig. 15). Two weeks after the final injection, larger blood samples were drawn for lymphocyte RNA extraction. Specifically, an Alpaca named Waikuri received injections of both OmpA-short and OmpF, while Thurbo was immunized with OmpA-long.

### Nanobody selections
Lymphocyte RNA extracted from immunized alpacas served as template for reverse transcription and subsequent amplification of the VHH/nanobody region[20] to generate a phagemid library. Two rounds of phage display were performed against purified and biotinylated OmpA-short, OmpA-long and OmpF in TBS, pH 7.5 and 0.03% β-DDM exactly as described elsewhere[30]. Phages were produced in *E. coli* SS320 Δ*ompA* or Δ*ompF* cells, which were generated by mating MC1061 Δ*ompA* or Δ*ompF* with XL1-blue. The second phage display selection resulted in a 764-fold, 7163-fold, and 1925-fold enrichment against OmpA-short, OmpA-long, or OmpF, respectively, as determined by qPCR compared to a control protein AcrB. From the enriched pools, 2000-2500 colony forming units (CFU) were subcloned into pNLx and nested with an ~30-fold excess of flycodes for flycode analysis[22].

### Deep sequencing and flycode assignment
Deep sequencing of the flycoded libraries was conducted following the previously described method[22]. In brief, flycoded nanobodies in the plasmid pNLx were excised and ligated with compatible double-stranded Illumina adapter oligonucleotides. The sample was then subjected to sequencing on an Illumina MiSeq Sequencer using a 600-cycle v3 MiSeq Reagent Kit, resulting in 2 × 300 base pair (bp) paired-end reads. Bioinformatic analysis of the deep-sequenced libraries using the previously published filtering steps[22] revealed the following unique nanobodies and flycodes: for OmpA-short, 1'040 unique nanobodies were nested with 29'746 unambiguous flycodes; for OmpF, 824 unique nanobodies were nested with 15'320 unambiguous flycodes; for OmpA-long, 1'739 unique nanobodies were nested with 35'718 unambiguous flycodes. Based on this analysis, a database for MS/MS ion search was generated, containing information about the association of each unique nanobody with its corresponding unambiguously assignable flycodes.

### Expression and selection of flycoded libraries
For flycode analyses, flycoded pools were produced in *E. coli* MC1061 carrying genomic deletions of *ompA* (for nanobodies raised against OmpA-short and OmpA-long) or *ompF* (for nanobodies raised against OmpF)[22]. Upon purification of the flycoded nanobodies via Ni-NTA chromatography, they were separated on a SRT SEC-300 column (Sepax) using PBS, pH 7.4 and the peak corresponding to monomeric nanobodies was pooled,

diluted to 0.02 mg/ml in PBS pH 7.4 supplemented with 0.5% BSA, and subsequently added to various bacterial strains.

For OmpA-short and OmpF, the flycoded nanobody pools were added to *E. coli* K-12 strain MC1061, the respective isogenic knockout strain along with a set of clinical strains (CS#1, CS#2, CS#3, and CS#4 all expressing the OmpA-short isoform, as well as CS#5, CS#6, CS#8 and CS#10 expressing OmpA-long). In the case of OmpA-long, lab strains MC1061 Δ*ompA-short::ompA-long* and the isogenic control MC1061 Δ*ompA-short::ompA-short* (both cases involved complementation by expressing the respective OmpA encoded on plasmid pBXNPH3) were used, along with clinical CS#8 and CS#11, both expressing OmpA-long.

50 ml cells with an $OD_{600nm}$ of 2 were harvested and washed once with 25 mL PBS pH 7.4, 0.5% BSA, and incubated for 20 min at room temperature before pelleting again. 25 ml diluted nanobody pools (0.65 mg, final concentration of 26 μg/mL; determined by measuring $A_{280}$ and assuming that 1 mg/ml flycoded nanobody has an $A_{280}$ = 5.0) were added to each cell pellet and incubated for 20 min at room temperature. Each cell pellet was washed three times with 25 ml PBS pH 7.4.

### Flycode extraction and purification
To isolate flycodes, we followed the previously established procedure[22]. Briefly, the cell pellet was solubilized in 25 ml 4.8 M guanidinium chloride. For quantification in MS/MS[22] a purified nanobody fused to 28 flycodes of known sequence[22] was spiked in and insoluble components were separated by centrifugation. Ni-NTA resin was incubated with His-tagged flycodes under denaturing conditions for 2 h at room temperature, and then transferred to a Mini Bio-Spin chromatography column (BioRad) and washed (3x 500 μL TH-Im buffer [20 mM TEAB, pH 8.0, 150 mM NaCl, 2.5 mM $CaCl_2$, 30 mM Imidazole, pH 8.0], and 2x 500 μL TH buffer [20 mM TEAB, pH 8.0, 150 mM NaCl, 2.5 mM $CaCl_2$]). The resin was incubated with 100 μL TH buffer containing 2.4 U thrombin (Merck; from human plasma) and incubated overnight at room temperature. The resin was washed (5x 500 μL TH-Im buffer) and the flycodes eluted with 250 μL TRY-Im buffer (20 mM TEAB, pH 8.0, 50 mM NaCl, 2.5 mM $CaCl_2$, 250 mM Imidazole, pH 8.0). The eluted flycodes underwent overnight digestion at 37 °C with 1 μg trypsin (2 μL of 0.5 ng/μL stock solution, Promega). The reaction was halted the following day by adding 20 μl of 5% (v/v) TFA. Subsequently, the sample was diluted with 250 μL 3% (v/v) ACN and 0.1% (v/v) TFA before being purified according to the StageTip protocol[59].

### LC-MS/MS analysis
For StageTip reverse-phase matrix preparation (using 3 M™ C18 Extraction Disks), the following steps were performed: Activation was achieved by applying 150 μL of 100% methanol, followed by pre-elution with 150 μL of a solution composed of 60% acetonitrile (ACN) and 0.1% trifluoroacetic acid (TFA). Subsequently, the resin was prepared for peptide binding using 150 μL of a solution consisting of 3% ACN and 0.1% TFA. Trypsin-digested samples (as described above) were diluted with an equal amount of 250 μL 3% ACN and 0.1% TFA and loaded onto the StageTips. After loading, the StageTips were washed three times with 150 μL of 3% ACN and 0.1% TFA.

Elution was carried out with 150 μL of 60% ACN and 0.1% TFA, followed by lyophilization. To reconstitute the flycodes, 15 μL of 3% ACN and 0.1% formic acid (FA) supplemented with indexed retention time (iRT) standard peptides (2xiRT kit, Biognosys) were added, and the samples were sonicated using a bath sonicator. Finally, 4 μL of the reconstituted samples were subjected to LC-MS/MS analysis. For OmpA-short and OmpF, sample analysis involved an ACQUITY M-class UPLC system (Waters AG) coupled with a Q-Exactive HF mass spectrometer (ThermoFisher). LC system equilibration used 99% solvent A (0.1% formic acid in water) and 1% solvent B (0.1% formic acid in ACN). Peptide trapping occurred on a Symmetry C18 trap column (5 μm, 180 μm × 20 mm, Waters AG) at a flow rate of 15 μL/min for 30 s. Subsequently, peptide separation utilized an HSS T3 C18 reverse-phase column (1.8 μm, 75 μm × 250 mm, Waters AG) with the following gradient: 8–20% solvent B in 60 min, 20–40% solvent B in 10 min, and 40−95% in 5 min. The flow rate remained constant at 0.3 μL/min, and

the temperature was maintained at 50 °C. Mass spectra were recorded in a data-dependent acquisition mode on a Q-Exactive HF mass spectrometer. MS1 spectra were acquired using a mass range of 350–1,500 $m/z$ at a resolution of 120'000 (at 200 $m/z$) with an automatic gain control (AGC) target of $3 \times 10^6$ and a maximum injection time of 50 ms. Peptide precursor with charge state between 2 and 7 were selected for fragmentation using quadrupole isolation (1.6 $m/z$ window), a resolution of 30'000 at 200 $m/z$, an AGC target value of $1 \times 10^5$ and a maximum injection time of 50 ms, with a normalized collision energy of 28%. Dynamic exclusion was activated and set to 15 s with a mass tolerance of 10 p.p.m.

For flycode analysis of nanobodies selected against OmpA-long, isolated and purified flycodes were analyzed using an ACQUITY M-class UPLC system (Waters AG) coupled to an Orbitrap Fusion Lumos Tribrid Mass Spectrometer (ThermoFisher). Trapping and elution of peptides were conducted as previously described. Acquisition of MS1 spectra was recorded in a data dependent mode using Orbitrap in the scan range of 300-1500 $m/z$, with an AGC target of $4 \times 10^5$, a resolution of 120,000 at 200 $m/z$, and a maximum injection time of 50 ms. For peptide precursors with charge states between 2 and 7, MS2 spectra were recorded with an IonTraputilizing a 1.6 $m/z$ isolation window, an AGC target value of 8000, and a maximum injection time of 80 ms. High-energy collisional dissociation fragmentation (HCD) was set to 30% collision energy. To prevent redundancy, dynamic exclusion was activated and configured with a 25-s interval, a mass tolerance of 10 p.p.m. A minimum signal intensity of 5000, and a maximum cycle time of 3 s was set.

## LC-MS/MS data processing
LC-MS/MS data was processed using Progenesis QI by Nonlinear Dynamics. Experiments with respective pools were aligned, and peaks with ion charges ranging from +2 to +4 were automatically identified. MS/MS fragment spectra, with a feature rank threshold of <5 and an ion fragment count limit of 1,000, were exported after applying deisotoping and charge deconvolution.

Mascot 2.5 (Matrix Science) was utilized to match MS/MS features to flycodes present in the respective databases. For all experiments, the database 'p1875_db8' was employed, as it contained the spiked standard, in addition to a Swiss-Prot database ('fgcz_swissprot_S') containing common contaminants and decoys. In the case of the OmpA-short and OmpF library, database 'p3127_db1' was used, and for OmpA-long, 'p3127_db6' was used. Scaffold (Proteome Software Inc.) was applied for protein identification from the Mascot search as described. The spectrum report from Scaffold was integrated into Progenesis QI. Within Progenesis QI, protein abundance was normalized using the spiked control nanobody NB-Control. Protein abundance was calculated by summing the MS1 intensities of each flycode associated with the respective nanobody. Binder candidates were selected based on high MS1 intensities, ratios of MS1 intensities between wild-type and knockout strains, and the number of flycodes associated with them.

## Nanobody expression and purification
Selected nanobody genes, codon-optimized for *E. coli*, were synthesized by Twist and cloned into pSBinit for purification[21,30]. The nanobodies were expressed in *E. coli* MC1061 knockout strains (Δ*ompA* or Δ*ompA*Δ*ompF* (double knockout)) depending on the target protein. Following periplasmic extraction, supernatants were subjected to Ni-NTA affinity chromatography (Qiagen) and SEC (SRT SEC-100 or SRT SEC-300, Sepax). In order to attach fluorophores (Alexa Fluor 647, AF647, Catalog #A20347, ThermoFisher or Dy-490 Maleimide, Catalog #490-03 Dyomics) or linkers (PEG$_{11}$-biotin, Catalog # 21911, ThermoFisher) using maleimide chemistry, a cysteine residue at the C-terminus of the nanobody was introduced by Quikchange mutagenesis using primer pair pSBinit_Cys_FW and RV. To avoid oxidation of the free thiol group, purification of these cysteine-containing nanobodies was performed in the presence of 2 mM DTT. After SEC, DTT was removed using a PD MidiTrap G-25 (28918008, Cytiva) desalting column equilibrated with degassed PBS, pH 7.0. Maleimide-functionalized AF647 or PEG$_{11}$-biotin, was added at 3.6-fold molar excess

and the reaction was carried out in maximally 1 mL for 1 h at 4 °C. Excess label was removed using another PD MidiTrap G-25 column, equilibrated with PBS, pH 7.4 h Labeling efficiency was assessed by absorbance, MS, and SDS-PAGE.

Nanobody constructs with peptide linkers were modified by replacing the myc-tag with either LEA (AEAAAKEAAAKEAAAKEAAAKA-LEAEAAAKEAAAKEAAAKEAAAKA), PAPA (GAAPAAAPAKQEAAA-PAPAAAKAEAPAAAPAAKA), or GS ((GGGGS)$_4$)-linker, followed by an Avi-tag for site-specific in vitro biotinylation with BirA and a 3 C cleavage site. Genes for these linkers were obtained from GeneUniversal. For constructs containing two binders, the nanobodies were separated by a flexible GS ((GGGGS)$_4$)-linker. Complete DNA sequences as well as expressed protein constructs are listed in Supplementary Fig. 15.

Nanobody constructs fused to a hIgG1 Fc part, codon-optimized for *homo sapiens*, were subcloned into pcDNA3.4-derived expression vectors under CMV promoter control, preceded by a mammalian Kozak sequence (GCCACC) and an immunoglobulin secretion signal (MDWTWRVFCLLAVAPGAHS), synthesized by GeneUniversal. Complete DNA sequences as well as expressed protein constructs are listed in Supplementary Fig. 15.

Transfection-grade plasmid DNA was prepared using a NucleoBond Xtra midiprep kit (Macherey-Nagel) and transfected into suspension Expi293 cells (ThermoFisher) using Expifectamine 293 transfection kit (ThermoFisher). Expression was carried out in 40−50 mL of 2–3 million cells/mL for 5 days at 37 °C in a humidified shaker maintained by 8% $CO_2$. Supernatants were collected, cleared by two centrifugation steps ($500 \times g$ for 5 min and $5000 \times g$ for 20 min), and then subjected to Protein A Agarose (Abcam) capture using 2–4 mL bead slurry for 2−4 h at room temperature h. The resin was transferred to a gravity flow column and washed with 50 mL PBS. Protein elution was performed with 0.1 M glycine, pH 3, and fractions were collected, rebuffered using 1/10 of the elution volume of 1 M Tris, pH 8.5, and concentrated with centrifugal spin filters with a MW cutoff of 100 kDa (Merck) and subjected for SEC using a Superose 6 column (Cytiva) after a centrifugal high spin of >20,000 $\times g$ for 10 min to remove potential aggregates. Purified HcAbs were finally run on Superose 6 column in PBS, and pure fractions, verified by SDS-PAGE, were pooled, concentrated, and stored at -80 °C after snap-freezing in liquid nitrogen until application.

Nanobodies for X-ray crystallography were cloned into pBXNPHM3 (Addgene: 110099)[21] and transformed into MC1061 Δ*ompA*. Nb01 and Nb39 were expressed at 20 °C overnight, harvested, and purified via Ni-NTA affinity chromatography (Qiagen) with subsequent 3 C protease cleavage and dialysis. Reverse Ni-NTA affinity chromatography was performed to remove 3 C protease, His-tagged MBP fusion protein and uncleaved protein, followed by concentration and SEC purification (SRT10C 300, Sepax) using TBS. Purified nanobodies were concentrated to >15 mg/mL.

## X-ray crystallography
The β-barrel domains of OmpA-short and OmpA-long were expressed and purified from plasmid p7X for tagless expression as described above. OMPs were concentrated to 20 mg/mL and mixed in a 1:1.1 molar ratio of OmpA to the respective targeting nanobody to a final concentration of 10 mg/mL. Protein mixtures were crystallized by the sitting drop vapor diffusion method at 20 °C. In case of OmpA-short in complex with Nb01, 100 nL of protein solution containing OmpA (10 mg/mL) and Nb01 (7.8 mg/mL) was mixed with 100 nL of reservoir solution (0.1 M sodium acetate pH 5.5, 0.2 M calcium acetate, 25% [w/v] PEG MME 2 K). The crystals were cryoprotected with cryoprotection solution (0.1 M sodium acetate pH 5.5, 0.2 M calcium acetate, 25% [w/v] PEG MME 2 K, 25% [w/v] glycerol) and then flash-frozen in liquid nitrogen. Diffraction data were measured at the beamline X06DA (PXIII) of the Swiss Light Source at a temperature of 100 K (Paul Scherrer Institute, Villigen, Switzerland) and processed using autoPROC[60] in the space group P21. Phases were obtained by molecular replacement using the Phaser module of the Phenix package using the transmembrane

domain of the OmpA PDB-ID: 1BXW and the Nb01 model generated by the AlphaFold2-based ColabFold as initial search model[61,62]. Two copies of the OmpA-Nb01 complex were present in the asymmetric unit. The model building was done manually in Coot[63]. The model was refined using phenix.refine module[64]. The crystals belonged to the $P2_1$ space group and contained two copies of the OmpA-Nb01 complex in the asymmetric unit.

For OmpA-long in complex with Nb39, 100 nL protein complex mixture of OmpA-long (10 mg/mL) and Nb39 (7.5 mg/mL) was mixed with equal amount of reservoir solution (0.1 M NaCl, 0.1 M Sodium citrate, pH 5.5, 12% PEG 4 K, 0.1 M LiSO4). Crystals were cryoprotected (0.1 M Sodium citrate, pH 5.5, 12% PEG 4 K, 0.1 M LiSO4, 30% glycerol) and snap-frozen in liquid nitrogen. Diffraction data were measured at the beamline X06SA (PXI) of the Swiss Light Source. The data were processed in the space group I222 using the Automatic Data Processing pipeline at the beamline[65]. The structure was solved by molecular replacement using Phaser module of Phenix using the OmpA-short model processed by Sculptor[66] and the Nb39 model generated by ColabFold. The model building was done in Coot and ISOLDE[67] and the model refinement was done using phenix.refine.

### Cellular binding assays

Unless otherwise specified, the indicated strains without an expression plasmid were cultured overnight in LB medium at 37 °C and 160 rpm. Strains containing plasmids for the expression of OmpA-long or OmpA-short (*E. coli* MC1061 Δ*ompA* with either pBXNPH3-ompA-short or pBXNPH3-ompA-long) were grown and induced with 0.05% L-arabinose according to the above-mentioned protein expression protocol. After expression and overnight growth, the $OD_{600 nm}$ was determined and adjusted to 1.0 in 1 mL PBS at pH 7.4. 1 mL cells of $OD_{600 nm}$ 1 were pelleted and washed in 500 μL PBS at pH 7.4 with 0.5% BSA. The cell pellets were then resuspended in 100 μL PBS with 0.5% BSA containing 1 μM AF647-labeled nanobody. The mixture was incubated for 20 min at room temperature. Unbound nanobodies were removed by washing the cells twice with 500 μL PBS. Finally, the cells were resuspended in 100 μL PBS, and fluorescence was measured (excitation: 650 nm; emission: 675 nm) and normalized to the $OD_{600 nm}$ using a plate reader (Cytation, BioTek).

For the detection of biotinylated or His-tagged nanobodies using Atto565-labeled streptavidin, washed cells were incubated with 100 μl PBS with 0.5% BSA containing 2.5 μM nanobodies for 20 min at room temperature. Unbound binders were removed by two washes with 500 μL PBS at pH 7.4 containing 0.5% BSA. Cell pellets labeled with biotinylated nanobodies were then resuspended in 100 μL 1 μM streptavidin-Atto565 in PBS at pH 7.4 with 0.5% BSA and incubated for 20 min. To detect nanobodies via the His-tag, previously coupled NTA-biotin-streptavidin-Atto565 dye was used. Two washes with 500 μL PBS removed unbound Streptavidin. In the final step, cells were resuspended in 100 μL of PBS, and fluorescence was measured (excitation: 563 nm; emission: 592 nm) and normalized to the $OD_{600 nm}$ using a plate reader (Cytation, BioTek).

### Analysis of Nb01 and Nb39 specificity by high-throughput flow cytometry

*E. coli* (28 representative strains from the Swiss database and 12 strains from official culture collections, see Supplementary Table 8) were cultured in Lactose Broth (LacB) overnight at 37 °C. Bacterial strains other than *E. coli* (see Supplementary Table 8), were cultured as recommended by the respective culture collection (DSMZ or ATCC).

To determine cell densities, overnight cultures were stained with 1x SYBR Green I (S9430-.5 ML, Sigma-Aldrich), 20 μg/mL propidium iodide (P1304MP, Molecular Probes) and quantified by flow cytometry using a CytoFLEX (Beckman Coulter), equipped with a 488 nm laser and filter sets of 525/40 (green channel) and 690/50 (red channel) at a flow rate of 100 μL/min. Based on this initial quantification, bacteria were diluted in PBS supplemented with 1 mM EDTA to ~100,000 cells per mL and stained for 1 h at room temperature with either 1 nM Dy490-labeled Nb01 or 1 nM Dy490-labeled Nb39 and 0.5 μg/mL propidium iodide. To obtain total cell counts independently of nanobody staining, the same bacterial dilutions were

separately stained with 1x SYBR Green I, 20 μg/mL propidium iodide for 1 h at room temperature. To test for Nb01 and Nb39 specificity, the stained samples were analyzed by flow cytometry as described above. For evaluation of Nb01 and Nb39 specificity, intact bacteria signal with a green intensity brighter than 2000 AU are quantified within a pre-defined gate and compared to the intact bacteria concentration of the SYBR Green reference. Strains with nanobody stained event counts >50% of the SYBR Green reference, were considered as detected (see Supplementary Fig. 7 and Supplementary Table 8). Strains with nanobody stained event counts <1% of the SYBR Green reference were considered not detected.

### Nanobody characterization using flow cytometry – sample preparation

To characterize nanobodies in detail by flow cytometry, the indicated *E. coli* strains including clinical isolates were cultured overnight in LB media or induced for protein expression as per the above described protocols when required. All reagents were sterile-filtered through a 0.22 μm pore filter.

To begin, 1 mL of culture was taken and washed in 0.85% NaCl or PBS, and the cell density was adjusted to $3 \times 10^7$ cells/mL assuming a concentration of 8 x $10^8$ cells/ml at $OD_{600 nm}$ of 1. In the initial step, 50 μL of cells were stained with the indicated amounts of Alexa Fluor 647-labeled nanobody or nanobody construct. If not mentioned otherwise, cells were fixed after staining. To this end, cells were washed twice in 0.85% NaCl and then resuspended in 2% PFA. The samples were incubated at 4 °C for 45 min in the dark. After incubation, PFA was removed through another washing step. Finally, cells were resuspended in a solution containing 3.34 μM Syto9 and 20 μM PI (from the LIVE/DEAD™ BacLight™ Bacterial Viability and Counting Kit, ThermoFisher) for viability staining. For samples without fixation, cells were washed twice with PBS, and ~150,000 cells were added to each well. 50 uL of labeled nanobody was added to the respectively indicated final concentration. Finally, 50 uL of Syto9 and PI were added to a final concentration of 3.34 μM and 20 μM, respectively.

For each measurement, single stains and unstained controls were recorded for each strain used. To obtain a PI-positive control, cells were incubated in 70% ethanol for 30 min and washed twice in 0.85% NaCl or PBS before adding PI.

### Nanobody characterization using flow cytometry – measurements

Data acquisition was performed using either a CytoFlex S (Beckman Coulter) or an Aurora (Cytek) flow cytometer, as indicated for the respective experiment. The Aurora flow cytometer seems to be more sensitive, collecting more signal, however, measurements on both cytometers results in a similar EC50 value (Supplementary Fig. 10c). The CytoFlex S is equipped with 405 nm, 488 nm, 561 nm, and 640 nm lasers, while the Aurora additionally features a 355 nm laser. For the CytoFlex S, emitted fluorescence light was collected with 525/40 nm (blue channel), 610/20 nm (yellow-green channel), and 676/19 nm (red channel) bandpass filters. The Aurora utilized virtual filters set at 508/20 nm (blue), 615/20 nm (yellow-green), and 679/18 nm (red). To minimize background noise, a threshold of 650 was applied for FSC, and 750 for SSC during measurements on the CytoFlex S. For the CytoFlex S, gains were set as follows: Syto9 = 100, PI = 600, AF647 = 750. For the Aurora, thresholds were adjusted within the range of 4000 - 6000 for FSC and 16000−21000 for SSC, without affecting the bacterial population. The gain settings for the Aurora were: FSC = 950, SSC = 735, B1 = 1200, YG3 = 920, R2 = 1500. In both cases, data was acquired at a flow rate of 100 μl/min. Data analysis was performed using R (v. 4.0.2) and FlowJo (v. 10). In R, we utilized the flowAI (v. 1.18.5), flowCore (v. 2.0.1), flowWorkspace (v. 4.0.6), ncdfFlow (v. 2.34.0), and flowStats (v. 4.0.0) packages for evaluation. All datasets underwent quality control using the flow_auto_qc() function from the flowAI package with default settings. Fluorescence intensity was shown using the determined earth mover's distance (EMD) instead of median fluorescence intensity (MFI). EMD quantifies biologically meaningful differences between a control sample, in our case a non-related control binder (SB-nr: a non-randomized synthetic nanobody

(sybody) from the concave library[21], and the investigated nanobody (Supplementary Fig. 10b)). MFI in contrast relies on normal distribution of the fluorescence intensity peaks and loses information about population distribution. To calculate the EMD, we use Sb-nr at a concentration of 33 nM and calculate its dissimilarity to different nanobody concentrations using the R package transport (v.012-2). Determined EMDs were plotted against nanobody concentration. Curves were fitted with GraphPad prism v9.3 and a variable slope model using the Hill equation to determine EC50 values. Standard deviations were determined from three EC50 values of biological replicates.

### LPS extraction and separation on Tricine-SDS-PAGE

LPS was extracted by resuspending the bacterial pellet at $OD_{600}$ of 10 in 1 mL distilled water. 25 µL bacterial suspension was mixed with 25 µL 2× sample buffer (200 mM Tris-HCl (pH 6.8), 40% glycerol, 2% SDS, 0.04% Coomassie Blue, and 2% β-mercaptoethanol) and incubated at 100 °C for 10 min. DNaseI was added to a final concentration of 100 µg/mL, followed by incubation at 37 °C for 30 min. Subsequently, proteinase K was introduced to a final concentration of 2 mg/mL, and the mixture was incubated at 60 °C for 1 h.

Tricine-SDS-PAGE was employed for the separation and analysis of the extracted LPS. A 20% acrylamide separation gel was prepared by mixing acrylamide/bisacrylamide (37.5:1, 40%), 3 M Tris-HCl (pH 8.5), SDS (20%), glycerol (50%), ammonium persulfate (APS, 10%), and N,N,N',N'-Tetramethylethylenediamine (TEMED). For the 4% acrylamide stacking gel, acrylamide/bisacrylamide (37.5:1, 40%), 3 M Tris-HCl (pH 8.5), and SDS (20%), were combined with APS (10%) and TEMED. LPS samples were diluted 1:5 with sample buffer and loaded onto the gel. Electrophoresis was carried out at 100 mA per gel for 1 h in migration buffer (100 mM Tris-HCl (pH 8.3), 100 mM Tricine, and 0.1% SDS). The gel was subsequently stained using Emerald stain according to manufacturer's instructions (P20495, ThermoFisher).

### Capture assay

Unless otherwise specified, E. coli strains were cultured in Luria Broth (LB, 6271000, LLG Labware). Where specified, LB supplemented with 0.5 mM EDTA, or Lactose Broth (LacB, 70142, Merck) were used. The capture assay was performed in two procedures. The first procedure involved preparing magnetic beads by coupling them with biotinylated nanobody constructs, followed by adding these prepared beads to the diluted or spiked bacteria. The second procedure entailed incubating the biotinylated constructs with the bacteria, diluting the solution, and then capturing the mixture with magnetic beads.

For procedure 1, cultured strains were diluted to ~500 CFU in 1 mL PBS unless stated otherwise, and then mixed with the prepared beads. To prepare the beads, 5 µg of magnetic streptavidin beads (Dynabeads™ MyOne™ Streptavidin C1) were washed three times in PBSTB (PBS, 0.05% Tween20, 1% BSA). The beads were then resuspended in 1 mL of PBS and coupled with biotinylated nanobody constructs added at a 1.5-fold molar excess to the maximal binding capacity of the magnetic beads. The reaction proceeded for 30 min at room temperature with end-to-end rotation. Unbound nanobody was removed by using a magnet, and unoccupied biotin binding sites were saturated by adding 1 mL of 5 mM biotin in PBST for 5 min. Subsequently, the beads were washed three times in PBSTB and used directly for capture.

In the case of procedure 2 of the capture assay, bacteria were diluted to an $OD_{600\,nm}$ of 0.01, and incubated with 65 nM of biotinylated nanobody constructs. According to literature, one E. coli cell has about 100'000 OmpA molecules per cell and at $OD_{600\,nm}$ of 1 a concentration of $8 \times 10^8$ cells/mL. Using the Avogadro constant, an E. coli culture of an $OD_{600\,nm}$ of 0.01 has an OmpA concentration of 1.32 nM. To reach an excess of about 50-fold nanobody over OmpA, 65 nM nanobody is required. The mixture was incubated for 1 h at room temperature while shaking. Cells were then diluted three times at a 1:10 ratio in PBSTB and combined with previously washed magnetic beads in PBSTB. The capture was performed for 20 min at room temperature with end-over-end rotation.

In both cases, after capture, the beads were separated using a magnet, and the supernatant was removed. A 1/10 fraction of the supernatant was plated on LB agar for CFU counting, and the beads were resuspended in a small volume of PBS for plating and CFU counting as well. Capture efficiencies were determined by dividing the bead-bound fraction (BBF) by the sum of the BBF and unbound fraction (UBF) counted in the supernatant.

### Sequence coverage and specificity determination

To study OmpA diversity in sequenced E. coli isolates, two databases were screened. First, 2,093 genomes derived from clinical isolates from the University Hospital Basel (USB) and the Institute of Medical Microbiology Zürich (IMM) were screened using blastn v2.13[68] and the ompA nucleotide sequence of K-12 (NC_000913.3). Gene sequences were translated to protein sequences, aligned using prank v.170427[69] and a tree was calculated using IQ-TREE v2.2.0.3[70] and midpoint-rooted. Sequences were further analyzed in R and the loop variants identified (defined as unique combinations of the extracellular OmpA loops 1 to 4). Four outliers due to missing data or extremely divergent sequences (Supplementary Fig. 1a) were excluded. The sequences of the complete region on the nucleotide level ranging from loop 1 to loop 4 of the clinical strains were then used to query the 661k database[28] using cobs v0.2.0[71] and a kmer similarity range from 0.8 to 1 to capture more diversity present in OmpA in E. coli strains. Hits were analyzed adapting the R script by Blackwell et al.[28]. The identified E. coli genomes were then used to build a database and queried with blastn using the reference ompA gene sequence as before. Sequence data was analyzed in R to identify loop variants as before. The individual loop sequences were concatenated and clustered using hierarchical clustering based on the Levenshtein distance in R. To identify possible cross-reactions of the nanobodies with other protein sequences similar to the extracellular ompA loops found in E. coli, hmmer profiles (hmmer v3.3.2[72]) were generated using the amino acid alignments for each loop and queried against all proteins of E. coli as well as all proteins of other bacterial species (RefSeq release 213[73]) with a sequence E value threshold of 0.001.

### Statistics and reproducibility

For the statistical analysis of EC50 values determined by flow cytometry, three binding experiments using independently inoculated bacterial cultures were carried out to determine mean and standard deviation. Statistical parameters that were part of bioinformatic analysis or X-ray structure determination were directly taken from the respective programs used to analyze the data (see also Reporting Summary). Cellular binding and capture assays were carried out as at least two biological replicates under identical experimental conditions and representative data of one experiment are shown.

### Reporting summary

Further information on research design is available in the Nature Portfolio Reporting Summary linked to this article.

### Data availability

The mass spectrometry proteomics data have been deposited to the ProteomeXchange Consortium via the PRIDE partner repository with the dataset identifier PXD054264. The custom software used to design the flycode library and to filter and analyze NGS data is available on https://github.com/cpanse/NestLink. Scripts of the bioinformatic analysis of ompA diversity are available at https://gitlab.uzh.ch/appliedmicrobiologyresearch/amr_publications/ompa_paper. Structural models have been deposited to the Protein Data Bank (PDB) with the accession codes 9FZC and 9FZD. Original raw data is found in the Supplementary Data file associated with this manuscript.

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

## Acknowledgements
We thank Saša Štefanić of the Nanobody Service Facility (NSF) at UZH for conducting alpaca immunizations and Beat Blattmann of the Protein Crystallization Center at UZH for setting up crystallization screens. We are also grateful to Martin Jinek and Kenny Jungfer for sharing their measurement time and assistance during sessions at the Paul Scherrer Institut (PSI). Stephan Benke at the Cytometry Facility at UZH is acknowledged for his invaluable assistance in setting up flow cytometry experiments, as well as for his guidance in the subsequent data evaluation. The authors gratefully acknowledge the Functional Genomics Center Zurich (FGCZ) of University of Zurich and ETH Zurich, and in particular Bernd Roschitzki, Christian Panse, and Lennart Opitz, for the support on Proteomics and Genomics analyses. We thank Tim Roloff for whole genome sequencing of strains. We thank Christophe Beloin from Institut Pasteur for sharing the LPS extraction protocol. This work was supported by a Swiss National Science Foundation grant (NRP72 program, number 407240_177368, to MAS and PMK), a Swiss National Science Foundation professorship (PP00P3_170625, to MAS) and a SNSF Bridge Discovery grant (40B2-0_187170, to MAS).

## Author contributions
M.A.S., P.M.K. and H.A.K. conceived the project and acquired funding. L.M.H. established the purification of OmpA and OmpF, prepared these proteins for Alpaca immunizations, carried out phage display, analyzed the flycode data and performed initial cellular binding assays to identify Nb01, Nb18 and Nb39. M.S. carried out parts of the flycode analyses under the supervision of L.M.H., established the labeling procedure with AF647 and PEG$_{11}$-maleimide, and performed the majority of the cellular binding assays. L.M.H. and M.S. generated gene deletions in *E. coli* MC1061. M.S. established and conducted all capture assays, planned and executed the linker engineering and purified all nanobody constructs with elongated linkers. F.A. and M.S. determined the crystal structures. M.M.R. established the flow cytometry analyses with the AF647-labeled nanobodies and A.P. as well as M.S. performed the flow cytometry analyses used to determine EC50 values. G.M. and D.M. performed high-throughput flow cytometry analyses to assess the species coverage and specificity of Nb01 and Nb39. M.S. performed the LPS analyses. F.W. and A.C. performed all bioinformatics analyses on OmpA. F.I. retrieved OmpA sequences from *E. coli* strains isolated at the Institute of Medical Microbiology and provided these strains. M.S. prepared the majority of the figures. F.W. and D.M. prepared figures.

M.S., L.M.H., D.M., M.M.R., H.A.K., A.E., P.M.K. and M.A.S. supervised research. M.S. and M.A.S. wrote the paper with major contributions from L.M.H., A.P., F.W., D.M., A.C., A.E. and P.M.K.

## Competing interests

A patent to protect the nanobody sequences has been applied (UZ513EP). M.S., L.M.H., P.M.K. and M.A.S. are listed as inventors. The flycode technology has been patented (WO2018078167A1). M.A.S. is listed as inventor. L.M.H. is an employee at Linkster Therapeutics AG. M.A.S. is co-founder, shareholder and board member of Linkster Therapeutics AG. D.M. and G.M. are employees of rqmicro AG. H.A.K. is co-founder, shareholder and CEO of rqmicro AG.
