## [Peer Review file · Communications Biology]

Rapid detection and capture of clinical *Escherichia coli* strains mediated by OmpA-targeting nanobodies

Corresponding Author: Professor Markus Seeger

This manuscript has previously been submitted to another journal. This document only contains information relating to versions considered at Communications Biology.

Version 0:

Reviewer comments:

Reviewer #1

(Remarks to the Author)

The manuscript by M. Sorgenfrei et al. presents an elegant study that isolates and characterizes the affinity, specificity, and crystal structure of nanobodies (Nbs) binding to two major isoforms of OmpA (long and short), which are found in over 90% of sequenced *E. coli* genomes. The authors convincingly demonstrate that the two best Nb candidates can recognize and label *E. coli* bacteria, both from lab and clinical strains, expressing these OmpA alleles. They also perform rigorous molecular engineering of these nanobodies, incorporating various constructs with distinct protein linkers to prevent the masking of the OmpA epitope by the O-antigen of LPS found in most clinical isolates. Several constructs were found to effectively label bacteria with LPS, allowing the recognition of OmpA and enabling a procedure to immunocapture bacteria from buffer solutions with low CFU/ml (50-500), within the range of *E. coli* bacteria found in blood samples from sepsis patients. In addition to Nbs binding OmpA, the authors also report anti-OmpF nanobodies, which are much less efficient for labeling and capturing bacteria, likely due to the lower expression levels of OmpF in *E. coli* compared to OmpA. This study is particularly interesting for protein and antibody engineers because of the Nb selection strategy that integrates proteomics and deep sequencing data (NextLink, as reported by these authors earlier) and the detailed characterization and engineering of Nbs for binding, labeling, and capturing *E. coli* cells with O-antigen LPS.

Major Issue

The study lacks relevant applications using actual clinical samples (e.g., blood or urine from patients) to demonstrate the advantage for early diagnosis compared to conventional subculturing. Additionally, the experimental translation of bacterial capturing to a faster phenotypic determination of antibiotic susceptibility is not shown, which is crucial for appropriate antibiotic treatment of patients. Although the authors discuss DNA sequencing of bacteria post-capture, this approach may not be practical or fast enough, as sepsis patients require almost immediate decisions in intensive care units within a few hours for a clinical impact. Capturing *E. coli* is an important step but should be coupled with fast phenotypic characterizations of the bacteria. This should be demonstrated with relevant clinical sample examples for publication in a journal with a broad scope such as Communications Biology. As it stands, the work seems more suitable for a specialized journal in protein engineering and antibody selection.

Minor Issues

1. The authors should show whether any Nbs from the panels selected against OmpA-long/short can bind to representatives of the OmpA-ND#1-5, which are non-detectable with the two main nanobodies reported in this work. If unreactive, a potential strategy to identify strains with these variants should be discussed.

2. Figures should be reorganized for clarity and ease of reading. Below are some suggestions:

-Figure 1 should focus solely on the selection and characterization of nanobodies. Although the original publication of the NextLink analysis is referred to, it would be helpful for the general reader to include a scheme of the methodology in Figure 1a, outlining the basic steps of the selection process rather than a scheme of OmpA and Nb.

- Figure 1b can be improved for clarity by adding the sample names below each column on the x-axis and using textures in the bars. Error bars and statistical information should also be included instead of using a large panel of similar colors, which can be confusing and inaccessible to color-blind readers.

- Panels 1c, 1d, and 1e can be moved to the supplementary section, whereas data shown in Tables S3 and S5 should be included in the main text.

3. Binding of Nbs to OmpA sequences in databases (Supplementary Fig. 2) should be moved to the main text.

4. The order and identity of Nbs used in the hcAbs with Nb repeats should be clearly indicated in both the text and figures.

Reviewer #2

(Remarks to the Author)

This is an interesting, albeit complex, study examining the ability of two nanobodies to bind to >90% of OmpA expressed by the majority of E. coli strains, independent of the expression of other antigens including O antigen type. A key focus of the study is the accessibility of OmpA in the presence of different levels of O-antigen. The ultimate aim is to be able to use these findings to generate a rapid diagnostic that can be used to recognise a broad range of E. coli strains rapidly. This study is an important milestone towards that. Although the occlusion conferred by O-antigen has been studied before I think this work brings a novel angle to its study. Thus, I think the study is overall exciting and in some ways considerably expands the field, and I really thought the crystallisation data examining the binding to the loops a valuable contribution. Some rationalisation and simplification would help.

Points to address

Do the authors have a view on why OmpA splits into short and long loop groups. Also, how are these defined?

No mention of K antigen is made. Do the authors consider this to play an occluding role and if not why not?

More detail on how the Flycode technology works here would be useful, albeit in an additional supplementary figure

Line 153-4 - How does this compare to an expected interface for a human or mouse Ab? This would help to understand the context of the findings

Line 220. ref missing and definition of what destabilises means here?

Line 230 Can't see the EDTA on the fig or in the legend

Line 251 would it be better to add bacteria to blood mouse etc to see how reproducible in a complex medium. Is this possible, or can discuss

Line 256 please show the raw data points as the spread is considerable

Line 267 does ref 35 refer to OmpF or OmpD/NmpC?

Line 270 How was Nb18 selected? What is the rationale for using this? I find this part confusing as it is used as a comparator but the evidence that it is similar is not entirely clear. Are the effects described only because of surface frequency of OmpF or is it not able to access OmpF equally well? What if OmpA and OmpF are jointly targeted?

Line 322-3, these conclusions are only valid with the nanobodies selected.

Supplemental tables – these seem overly complicated, can they be rationalised?

Supp Fig 1 The frequency should be presented as a log scale not as 10 000s

Supp Fig 7, should read exemplar

Supp Fig 8 – better labelling of Loop regions on the figure would be helpful

Reviewer #3

(Remarks to the Author)

Dear,

This study identified nanobodies (Nb) specifically targeting E. coli bacteria through binding to OmpA/F. This is a well-written manuscript that uses dynamic and logical flow. I enjoyed reading it. The authors nicely explain the field's current state of the

art and the limitations their study might overcome. They used adequate controls when appropriate, relevant biological conditions (CFUs/ml of medium, clinical isolates). They even generate new fundamental data (such as carbon sources and O-antigen density) for a broadly used bacterial lab model. They also used structural biology to map the epitope bound. The advantages of using Nbs are also highlighted in their manuscript. This comprehensive study generates significant support for further targeted therapies and diagnosis purposes.

Line 94: Can the authors provide references for the abundance of OmpA in *E. coli*, also for other bacterial species (as people might use that strategy to target other bacteria by generating new Nbs), compared to OmpF from the literature? What is known about that topic in a more precise and quantitative way?

Were the different binding abilities of all the Nbs tested using BLI/Octet? Using purified protein and/or using whole *E. coli* cells? Line 299: was the affinity of that Nb tested biochemically? The fusion itself might change its binding affinity.

Is the capsule/abundance/thickness of some *E. coli* isolates a problem for Nb/target recognition?

Line 296 "bulky": Is there any semi-quantitative way of giving more details concerning the volume/size of the proteins/fusions mentioned? Alpha fold might be helpful in this context as well. What is the critical size/volume ?

It might have been good to generate fluorescence microscopy pictures of the fluorescently labeled Nbs deposition on the bacterial cell surfaces to show that a membrane labeling following the OmpA/F pattern is followed, when appropriate. This could inform about the homogeneity of the bacterial labeling at the single-cell level using a direct visualization method.

Version 1:

Reviewer comments:

Reviewer #1

(Remarks to the Author)

The revised manuscript has addressed most points raised in my first revision. Although detection of *E. coli* in clinical samples has not been conducted, I understand this could be beyond the scope of this work. However, since the work is limited to efficiently capture *E. coli* bacteria from growth media and not from clinical samples or other complex samples, I recommend to remove "detection" from the title of this paper, since it implies to the reader that detection of *E. coli* is done from complex samples where its presence is not known, which is not the case from standard cultures of strains.

Reviewer #2

(Remarks to the Author)

All my queries are addressed. I am supportive of acceptance of this excellent study

Reviewer #3

(Remarks to the Author)

Dear,

The authors answered my comments, and I can witness that the manuscript and the whole study are strengthened.

We thank the three reviewers for carefully assessing our extensive study on OmpA-specific nanobodies. We found the comments to be very helpful to revise and further strengthen our manuscript. Addressing the concern that the *E. coli* capsule might provide an additional barrier that might impede nanobody binding, we have included additional data showing that a well characterized *E. coli* K1 strain containing a dense capsule can be stained with nanobodies akin to the clinical strains we have used in this studies. The point-by-point answers to the reviewer questions can be found below.

Reviewers' comments:

Reviewer #1 (Remarks to the Author):

The manuscript by M. Sorgenfrei et al. presents an elegant study that isolates and characterizes the affinity, specificity, and crystal structure of nanobodies (Nbs) binding to two major isoforms of OmpA (long and short), which are found in over 90% of sequenced *E. coli* genomes. The authors convincingly demonstrate that the two best Nb candidates can recognize and label *E. coli* bacteria, both from lab and clinical strains, expressing these OmpA alleles. They also perform rigorous molecular engineering of these nanobodies, incorporating various constructs with distinct protein linkers to prevent the masking of the OmpA epitope by the O-antigen of LPS found in most clinical isolates. Several constructs were found to effectively label bacteria with LPS, allowing the recognition of OmpA and enabling a procedure to immunocapture bacteria from buffer solutions with low CFU/ml (50-500), within the range of *E. coli* bacteria found in blood samples from sepsis patients. In addition to Nbs binding OmpA, the authors also report anti-OmpF nanobodies, which are much less efficient for labeling and capturing bacteria, likely due to the lower expression levels of OmpF in *E. coli* compared to OmpA. This study is particularly interesting for protein and antibody engineers because of the Nb selection strategy that integrates proteomics and deep sequencing data (NextLink, as reported by these authors earlier) and the detailed characterization and engineering of Nbs for binding, labeling, and capturing *E. coli* cells with O-antigen LPS.

We thank reviewer#1 for this positive assessment of our study.

Major Issue

The study lacks relevant applications using actual clinical samples (e.g., blood or urine from patients) to demonstrate the advantage for early diagnosis compared to conventional subculturing. Additionally, the experimental translation of bacterial capturing to a faster phenotypic determination of antibiotic susceptibility is not shown, which is crucial for appropriate antibiotic treatment of patients. Although the authors discuss DNA sequencing of bacteria post-capture, this approach may

not be practical or fast enough, as sepsis patients require almost immediate decisions in intensive care units within a few hours for a clinical impact. Capturing E. coli is an important step but should be coupled with fast phenotypic characterizations of the bacteria. This should be demonstrated with relevant clinical sample examples for publication in a journal with a broad scope such as Communications Biology. As it stands, the work seems more suitable for a specialized journal in protein engineering and antibody selection.

We agree with reviewer#1 that additional experiments that would demonstrate the successful application of our nanobodies for the rapid capture and diagnostics in a clinical setting would have been interesting. However, a conservative estimate of the time needed to perform such experiments (including the ethical approvals needed to perform such clinical validation studies) is clearly more than one year. Considering the extensive and complex results we report on in this paper, we feel that such a validation is out of scope.

Minor Issues

1. The authors should show whether any Nbs from the panels selected against OmpA-long/short can bind to representatives of the OmpA-ND#1-5, which are non-detectable with the two main nanobodies reported in this work. If unreactive, a potential strategy to identify strains with these variants should be discussed.

According to our in-depth NestLink experiments, we have evidence that the sequence space (and in particular the CDR3 sequences) of our nanobody pools are quite similar to each other. Hence, we expect that in the current set of available nanobodies, it is rather unlikely to find binder candidates that cross-react to the “non-detected” set of OmpA variants. The strategy we therefore currently follow-up is to generate new immunizations with OmpA-ND#1-5, and to perform additional screens to find additional nanobodies that close these gaps.

We mention this in our discussion:

Line 424: By analyzing the OmpA sequences that escaped detection by Nb01 and Nb39, we anticipate that another 3 – 4 nanobodies are required to achieve a coverage of close to 100 % of the strains producing OmpA.

2. Figures should be reorganized for clarity and ease of reading. Below are some suggestions:

-Figure 1 should focus solely on the selection and characterization of nanobodies. Although the original publication of the NextLink analysis is referred to, it would be helpful for the general reader to include a scheme of the methodology in Figure 1a, outlining the basic steps of the selection process rather than a scheme of OmpA and Nb.

We thank reviewer#1 for this helpful suggestion and we have generated an overview figure outlining the nanobody generation strategy as new Figure 1.

- Figure 1b can be improved for clarity by adding the sample names below each column on the x-axis and using textures in the bars. Error bars and statistical information should also be included instead of using a large panel of similar colors, which can be confusing and inaccessible to color-blind readers.

We have revised Fig. 1b (now Fig. 2b) to increase clarity and readability. Because the data was a “yes/no”-answer that was assessed in a larger screen at the onset of the project, these were single reads, this being the reason that we do not show any statistics here.

- Panels 1c, 1d, and 1e can be moved to the supplementary section, whereas data shown in Tables S3 and S5 should be included in the main text.

We would like to keep Fig. 1c-e in the main text (in the revised version Fig 2c-e), as they summarize the species coverage of the two nanobodies we identified in a general manner. Species coverage is of key importance to our study. Further, we do not think that the flycode-based ranking of binder candidates shown in Table S3 and Table S5 add value to the main body of the paper, because for each of these panels, only the “top binders” (i.e. Nb01 Nb39) were later characterized and discussed in the paper.

3. Binding of Nbs to OmpA sequences in databases (Supplementary Fig. 2) should be moved to the main text.

We consider Supplementary Fig. 2 as too complex to be shown as a main figure. Instead, we propose to keep Fig. 2c-e as main figures, because they aggregate the complex data shown in Supplementary Fig. 2 in a comprehensible fashion.

4. The order and identity of Nbs used in the hcAbs with Nb repeats should be clearly indicated in both the text and figures.

We thank the reviewer for this valuable suggestion. We have added the identities of the Nb units comprised in the hcAb construct in the figure legend of Fig. 6.

Reviewer #2 (Remarks to the Author):

This is an interesting, albeit complex, study examining the ability of two nanobodies to bind to >90% of OmpA expressed by the majority of *E. coli* strains, independent of the expression of other antigens including O antigen type. A key focus of the study is the accessibility of OmpA in the presence of different levels of O-antigen. The ultimate aim is to be able to use these findings to generate a rapid diagnostic that can be used to recognise a broad range of *E. coli* strains rapidly. This study is an important milestone towards that. Although the occlusion conferred by O-antigen has been studied before I think this work brings a novel angle to its study. Thus, I think the study is overall exciting and in some ways considerably expands the field, and I really thought the crystallisation data examining the binding to the loops a valuable contribution. Some rationalisation and simplification would help.

We thank reviewer#2 for this positive assessment of our work.

Points to address

Do the authors have a view on why OmpA splits into short and long loop groups. Also, how are these defined?

We thank the reviewer for this interesting question. As we mention in line 104, OmpA is the receptor of some *E. coli* bacteriophages. It has been previously proposed that the OmpA isoforms have evolved to evade phage infections, which in our view is a plausible explanation.

No mention of K antigen is made. Do the authors consider this to play an occluding role and if not why not?

We thank reviewer#2 for this important question. In our understanding of the literature, the O-antigen layer is more dense than the K-antigen layer and hence we mainly focussed our discussion on the O-antigens.

To address the reviewer's question, we show novel data in the revised version in which we assessed nanobody staining by FACS of the *E. coli* K1 (ATTC 11775) strain, which is the best described *E. coli* strain containing the dense K1 capsule. Of note, this strain also contains an O-antigen layer, and its OmpA-short sequence is identical in the loop regions as our reference OmpA-short sequence against which we generated Nb01. As can be seen in Fig. 4c, the observed EC50 of around 85 nM is in the same range as observed for clinical strain #2 (EC50 of around 60 nM) and the maximal binding signal is almost as strong as in the *E. coli* MC1061 strain devoid of capsule and O-antigen layer.

Our data clearly shows that an *E. coli* strain containing a thick K1 capsule can be recognized by our nanobody Nb01. Unfortunately, we did not have an isogenic strain at hand devoid of the K1 capsule to directly assess the contribution of the capsule to nanobody accessibility to OmpA. Further, we did

not assess whether the clinical strains we have worked with do contain a capsule or not, and if so, which one. We refrained from a bioinformatics analysis to assess the presence of the capsule, because in our view experiments are required to really show the presence and (more importantly) the abundance/thickness of the capsule.

More detail on how the Flycode technology works here would be useful, albeit in an additional supplementary figure

Following reviewer#1's suggestion, we made a new Figure 1 describing the technical workflow of our nanobody identification technology, which includes the flycode technology.

Line 153-4 - How does this compare to an expected interface for a human or mouse Ab? This would help to understand the context of the findings

A recent study on a larger number of SARS-CoV-2 nanobodies and antibodies revealed that the mean buried surface area of antibodies is around 100 Å² larger than that of nanobodies (PMID: 34344900).

We thus added in line 161:

A recent analysis of a larger number of SARS-CoV-2 nanobodies and antibodies revealed that the buried surface areas of nanobodies range from 500 Å² to 1000 Å² with a median of around 800 Å², indicating the binder interfaces of Nb01 and Nb39 are in the lower range of typical nanobody-target protein complexes. The same study revealed that the median buried surface area of antibodies is around 100 Å² larger than that of nanobodies.

What is, however, more relevant in the context of this study that the overall "footprint" of an antibody versus a nanobody is approximately twice as large. In other words, antibodies more likely experience steric clashes with O-antigens, which might prevent antibody binding to OmpA.

Line 220. ref missing and definition of what destabilises means here?

EDTA is known since very long to destabilize the outer membrane of Gram-negative bacteria by chelating divalent cations. We now cite this original work from the 1960ies (line 232 in revised manuscript):

Leive L, Shovlin VK, Mergenhagen SE. Physical, chemical, and immunological properties of lipopolysaccharide released from Escherichia coli by ethylenediaminetetraacetate. J Biol Chem. 1968 Dec 25;243(24):6384-91. PMID: 4973230.

Line 230 Can't see the EDTA on the fig or in the legend

We thank reviewer#2 for spotting this error. In earlier versions of Fig. 4, we also showed a lane where EDTA was added to the bacteria during growth. However, EDTA did not have a clear effect that would have been visible by this type of LPS analysis, in contrast to lactose broth having a very clear effect. The likely reason is that EDTA renders the O-antigens of the LPS more flexible/fluid by complexing divalent cations that bridge the sugar chains via electrostatic interactions. But it probably does not change the length or the density of the LPS itself.

We deleted the erroneous statement.

Line 251 would it be better to add bacteria to blood mouse etc to see how reproducible in a complex medium. Is this possible, or can discuss

We thank reviewer#2 for this valuable remark. Indeed, we tried spiking bacteria into human blood. Capture indeed worked efficiently from blood. However, we were faced with the problem of unspecific capture (i.e. bacteria lacking OmpA or carrying the alternative isoform, OmpA_short or OmpA_long, respectively) as well as with the challenge of reproducing data. Another problem was that the lab *E. coli* strains did not survive in blood, which would have allowed to carry out experiments with isogenic *ompA* deletions strains. Since the punchline of our paper is the specific detection and capture, we did not further invest in protocol optimizations that would have been necessary to overcome these problems.

Line 256 please show the raw data points as the spread is considerable

Thanks a lot for this suggestion. We replotted both Fig. 6c and 6d and show individual data points.

Line 267 does ref 35 refer to OmpF or OmpD/NmpC?

We thank the reviewer for spotting this error. Indeed, it is OmpD and not OmpF. Nevertheless, the meaning of our statements is still correct: akin to OmpF, OmpD is a trimer and has a large footprint. We corrected the error in the paper.

Line 270 How was Nb18 selected? What is the rationale for using this? I find this part confusing as it is used as a comparator but the evidence that it is similar is not entirely clear. Are the effects described only because of surface frequency of OmpF or is it not able to access OmpF equally well? What if OmpA and OmpF are jointly targeted?

As we describe in the section, our rationale was to use a OMP target with a larger footprint at the surface than OmpA. Since OmpF has twice as many beta-strands than OmpA and in addition is a trimer, it for certain has a larger footprint. As we point out, we were hoping that capture in clinical strains might work better with OmpF, owing to better accessibility. A further idea was also to co-target OmpA and OmpF with bi-paratopic constructs to increase the species coverage. However, it finally turned out that capture via OmpF was actually less efficient than via OmpA (probably owing to

the fact that there is around 10 times less OmpF at the cell surface). For staining, OmpF gave simply a much weaker signal and hence was for this reason less attractive than OmpA.

Hence, we agree with the reviewer that the OmpF part of the study can be seen as an unproductive detour. Nevertheless, the identified nanobodies against OmpF are *per se* of high quality and might be useful for the research community for staining or targeting purposes, this being the reason we wish to report them in this manuscript. If we do not publish them here, there is a high risk that we will never publish them.

Line 322-3, these conclusions are only valid with the nanobodies selected.

We tamed our statement saying that this are the conclusions we obtained with our studied set of nanobodies.

The section has been rephrased the following way (line 347)

Based on our findings with the nanobodies we have analyzed as part of this study, we concluded that i) OmpA is better suited as a target to stain and capture *E. coli* than OmpF, ii) structured and stiff linkers are potentially suited to bridge the LPS layer of clinical *E. coli* strains featuring dense O-antigen decoration as found in CS#2 and iii) the linker length is an important parameter if one attempts to capture clinical strains.

Supplemental tables – these seem overly complicated, can they be rationalised?

We understand the concern and this is why we have them added as supplementary tables. We have added these tables in order to comply with the requirements of the journal to give access to the full datasets and information.

Supp Fig 1 The frequency should be presented as a log scale not as 10 000s

Figure S1 does not contain frequencies. We looked in all our figures where larger numbers appear on the Y-axis and wish to keep it the way it is. This especially the case for Fig. 2c and d.

Supp Fig 7, should read exemplar

Changed.

Supp Fig 8 – better labelling of Loop regions on the figure would be helpful

Fig. S8 only shows loop regions, as explained in the legend (and obvious based on the interspersed regions and the low number of residues shown).

Reviewer #3 (Remarks to the Author):

Dear,

This study identified nanobodies (Nb) specifically targeting E. coli bacteria through binding to OmpA/F. This is a well-written manuscript that uses dynamic and logical flux. I enjoyed reading it. The authors nicely explain the field's current state of the art and the limitations their study might overcome. They used adequate controls when appropriate, relevant biological conditions (CFUs/ml of medium, clinical isolates). They even generate new fundamental data (such as carbon sources and O-antigen density) for a broadly used bacterial lab model. They also used structural biology to map the epitope bound. The advantages of using Nbs are also highlighted in their manuscript. This comprehensive study generates significant support for further targeted therapies and diagnosis purposes.

We thank reviewer#3 for this positive assessment of our research study.

Line 94: Can the authors provide references for the abundance of OmpA in E. coli, also for other bacterial species (as people might use that strategy to target other bacteria by generating new Nbs), compared to OmpF from the literature? What is known about that topic in a more precise and quantitative way?

There are some older papers wherein 2D-SDS-PAGE gels combined with proteomics were used to assess the expression level of OMPs in Gram-negative bacteria. However, the data between different papers is rather difficult to compare. Proteomic analysis would be more informative, but we did not find a paper that would provide comparable data between several Gram-negative bacteria under identical/similar conditions. Hence, we ourselves would suggest to perform a whole proteome analysis of the bacterial strains in order to obtain an idea of the abundance of target OMPs. We would combine proteomics data with bioinformatics analysis as performed in this paper to assess the sequence conservation of the OMPs. However, establishing a full proteomics protocol suitable for comprehensive detection and analysis of membrane proteins is out of scope for this work.

Were the different binding abilities of all the Nbs tested using BLI/Octet? Using purified protein and/or using whole E. coli cells? Line 299: was the affinity of that Nb tested biochemically? The fusion itself might change its binding affinity.

We thank the reviewer for this question. Curiously, we realized that Nb01 does not bind purified OmpA-short strongly, as opposed to the cellular context, where tight binding was observed. Since NestLink was used to score for high binding affinity in the cellular context, the lack of strong binding in detergent can be explained. It is not uncommon that binding affinities against a detergent-purified protein drastically differs from the affinity measured in the cellular context. BLI (or other)

experiments would have mainly assessed the affinity against detergent-purified OmpA, which is not meaningful in the context of this study.

Instead, we performed the titration experiments using flow cytometry to estimate apparent EC50 values, which are in the single digit nM range and thus very strong.

Finally, we did not determine EC50 values by flow cytometry for any of the fusion constructs. There were two reasons for this. Firstly, we were quite satisfied with the strong EC50 values observed for the individual nanobodies, and no further improvement seemed necessary. Secondly, we very likely would have run into the problem that the OmpA/OmpF target concentration (in the cellular context) could not have been further lowered due to signal detection reasons in flow cytometry in order to measure stronger affinities of the fusion constructs.

Is the capsule/abundance/thickness of some *E. coli* isolates a problem for Nb/target recognition?

We thank the reviewer for this valuable question. In the revised version of the manuscript, we added FACS titration data using the well-characterized capsulated *E. coli* K1 strain (see also answer to reviewer#2 to a similar question). Indeed, Nb01 can stain capsulated *E. coli* K1 strain efficiently (Fig. 4c).

Line 296 "bulky": Is there any semi-quantitative way of giving more details concerning the volume/size of the proteins/fusions mentioned? Alpha fold might be helpful in this context as well. What is the critical size/volume ?

That's a good point indeed. The molecular weight of the streptavidin tetramer is 55 kDa, which is around 3.5-times heavier than a nanobody (15 kDa) and 40-times heavier than then Alexa Fluor™ 647 dye (1.25 kDa). In terms of dimensions, a nanobody is around 2x2x4 nm³ and the streptavidin tetramer is around 4.5x5x5 nm³.

The text reads now as follows (line 319):

Avi-tagged nanobodies were then purified and enzymatically biotinylated to be recognized by Atto565-labeled streptavidin. With a dimension of around 4.5 x 5 x 5 nm³, the streptavidin tetramer is much larger than the AF647 dye and prevents the nanobodies (having itself a dimension of around 2 x 2 x 4 nm³) from binding to OmpA in clinical strains due to steric hindrance as opposed to nanobodies labeled with AF647 (Supplementary Fig. 5).

It might have been good to generate fluorescence microscopy pictures of the fluorescently labeled Nbs deposition on the bacterial cell surfaces to show that a membrane labeling following the OmpA/F pattern is followed, when appropriate. This could inform about the homogeneity of the bacterial labeling at the single-cell level using a direct visualization method.

We thank the reviewer for this interesting suggestion, which we will follow up in a future study. In

the context of this study (having respective controls such as the OmpA deletions strains), we provide sufficient evidence for surface staining. But what would be interesting to see is whether OmpA and OmpF are uniformly distributed at the cell surface, or cluster in some regions of the (growing and dividing) cell.